# A framework for integrating directed and undirected annotations to build explanatory models of cis-eQTL data

David Lamparter[¤a], Rajat Bhatnagar, Katja Hebestreit, T. Grant Belgard[¤b], Alice Zhang, Victor Hanson-Smith*

Verge Genomics, South San Francisco, CA 94080, USA

¤a Current address: Health 2030 Genome Center, 1202 Geneva, Switzerland
¤b Current address: The Bioinformatics CRO, Niceville, FL 32578, USA
* victor@vergegenomics.com

**Data Availability Statement:** Software package is available on github (https://github.com/dlampart/bagea/).

## Abstract

A longstanding goal of regulatory genetics is to understand how variants in genome sequences lead to changes in gene expression. Here we present a method named Bayesian Annotation Guided eQTL Analysis (BAGEA), a variational Bayes framework to model cis-eQTLs using directed and undirected genomic annotations. We used *BAGEA* to integrate directed genomic annotations with eQTL summary statistics from tissues of various origins. This analysis revealed epigenetic marks that are relevant for gene expression in different tissues and cell types. We estimated the predictive power of the models that were fitted based on directed genomic annotations. This analysis showed that, depending on the underlying eQTL data used, the directed genomic annotations could predict up to 1.5% of the variance observed in the expression of genes with top nominal eQTL association p-values < $10^{-7}$. For genes with estimated effect sizes in the top 25% quantile, up to 5% of the expression variance could be predicted. Based on our results, we recommend the use of *BAGEA* for the analysis of cis-eQTL data to reveal annotations relevant to expression biology.

## Author summary

Many geneticists wish to map changes in DNA sequences to changes in human traits and to understand these processes mechanistically. Here we present *BAGEA*, a framework to study this question for gene expression. Specifically, *BAGEA* models a genome variant's impact on gene expression based on established genome annotations. *BAGEA* predicts not only whether a variant has an impact on gene expression, but also the sign of the effect. We applied *BAGEA* to datasets from different tissues and cell types and found that annotations most predictive of gene expression in a given tissue were typically derived from similar tissues. Based on our results, we recommend the use of *BAGEA* to reveal annotations relevant to expression biology and to build predictive models of gene expression.

**Funding:** This research was supported by Verge Genomics, a venture funded drug discovery company (https://www.vergegenomics.com/). The funders of Verge Genomics had no role in study design, data collection and analysis, decision to publish, or preparation of the manuscript.

**Competing interests:** The authors of this paper are current and former employees of Verge Genomics, a venture-backed startup company. The authors have declared that no competing interests exist. This does not alter our adherence to all PLOS Computational Biology policies on sharing data and materials.

## Introduction

A longstanding goal in the field of genetics is to accurately predict the phenotypic consequences of any given variant from the genome sequence alone, i.e. to 'read the genome' [1]. This would help to reveal the phenotypic effects of very rare variants even if their effect is weak. The effects of such variants are typically studied via whole genome sequencing studies. However these studies often have limited statistical power because, by definition, there are few carriers in any sampled population [2].

Recently, progress has been made in predicting epigenetic marks and transcription factor (TF) binding from genome sequence alone; these sequence-based models predict the effect of any given sequence variant on epigenetic marks (and TF binding) [3–7]. The question now is how to extend these models to predict effects on genetically complex phenotypes, such as common diseases. A mechanistic stepping stone between the regulation of epigenetic marks and the regulation of complex phenotypes is the regulation of gene expression, as suggested by the previous observation that disease-causing sequence variants are enriched in gene expression quantitative trait loci (eQTLs) [8, 9]. Thus, there is a need for sequence-based models to predict gene expression.

One strategy to build sequence-based models of gene expression is to leverage sequence-based models of epigenetic marks. Results of these sequence-based models can be interpreted as *directed genome annotations*. A genome annotation is defined as a collection of genome regions that have a shared property such as coverage by a particular epigenetic mark, or evolutionary conservation across species. Each region can potentially carry an intensity value to denote the annotation strength, such as the strength of conservation. We call such an annotation *undirected* if its value is independent of the alleles its covering in a given individual. For *directed annotations*, the sign of its intensity value depends on characteristics of the sequence in the region, such as the presence of a specific allele. A simple motivating example is that of a SNP in a TF binding site. In this situation, the TF can have higher binding affinity for one allele versus the other allele. This can cause consistent directional transcriptional effects: the allele inhibiting binding of an activating TF for instance should lead to decreased expression of the target gene. Conversely, an allele inhibiting binding of a repressive TF would lead to increase in expression, allowing us to discern activators and repressors *de novo*. One strategy to express this effect as a directional annotation would be to use TF position weight matrices that calculate TF affinity for a given sequence, while computationally more sophisticated methods express the same relationship using deep neural networks [3–7].

Methods to evaluate the effect of directed genome annotations on gene expression have recently been proposed [7, 10]. Specifically, *Zhou et al.* predicted variant impact without exploiting eQTL data using models that predict expression from chromatin patterns directly [7]. *Reshef et al.* presented a fast method to determine which directed annotations are enriched in variants causal for a given phenotype. However, the method from *Reshef et al.* is geared towards screening and hypothesis testing rather than towards detailed predictive modeling. For instance, the Reshef model does not account for interactions between the effect of an annotation and the distance to the transcription start site (*TSS*).

Approaches using directed annotations to predict gene expression have been developed relatively recently. Methods integrating undirected annotations with eQTL data have a longer history. These methods allow the prior probability distribution of a SNP's effect size to vary based on the genome annotations with which it overlaps. This is achieved via bayesian hierarchical models [11–15]. This in turn allows to fine-map causal SNPs, find annotations that are either enriched or depleted in causal SNPs, and increase power to call eQTLs. Methods differ in the modeling assumption they make. For instance, assuming only one causal SNP in a locus

makes the model independent of linkage disequilibrium (LD), thereby simplifying modeling approaches and lower computational burden [11, 12, 15]. Allowing for multiple causal SNPs per locus can improve fine mapping but necessitates the modeling of LD [13, 14]. These types of models have also been employed for integrating functional annotations with GWAS signal [14, 16].

Here we present a new predictive model of gene expression, named Bayesian Annotation Guided eQTL Analysis (*BAGEA*). *BAGEA* is a variational Bayes modeling framework to analyze eQTLs using both directed and undirected annotations in a multivariate fashion. *BAGEA* can model interactions between these annotations by weighting the impact of the directed annotation based on the undirected annotations. Consequently, *BAGEA* can directly model phenomena relevant to genetic architecture, such as the relatively larger impact of SNPs close to the TSS on directed annotations compared to that of distal SNPs, making *BAGEA* mores useful for predictive modeling. *BAGEA*'s results are interpretable and highlight genome annotations that are particularly predictive for gene expression. Further, *BAGEA* can model multiple causal SNPs per region. Our software implementation of *BAGEA* can be run on summary statistics using external LD information as well as on individual level genotype data directly. Optionally, using a low rank approximation of the LD information improves run-time and decreases *BAGEA*'s memory requirements.

We used *BAGEA* to analyze results from a *cis*-eQTL meta-analysis in human monocytes and from *cis*-eQTL summary statistics derived from tissues of various origins [9, 17]. As additional input, we gave the method regulatory impact predictions of common variants on epigenetic marks from a recent deep neural network model [7]. We specified these predictions as directed annotations in the method. We show that *BAGEA* highlighted biologically sensible annotations as particularly predictive of eQTLs. Further we estimated the predictive power of the directed annotations for various eQTL datasets. Overall, our results suggest that *BAGEA* is a useful framework to build predictive models of gene expression based on directed annotations, find biologically relevant annotations, and benchmark methods that produce such directed annotations.

## Model overview

*BAGEA* models gene expression as dependent on SNP genotypes in *cis*. In general, SNP effects on gene expression depend on both directed and undirected annotations (Fig 1A). *BAGEA* builds predictors of gene expression and ranks annotations by their impact on gene expression. For every gene $j$, *BAGEA* takes as input a genotype matrix $X_j$, an expression vector $y_j$, annotation matrices $V^j$, $F^j$ and $C_j$. $X_j$ has dimensions $(n \times m_j)$, where $n$ is the number of individuals assayed, and $m_j$ is the number of SNPs in *cis* of gene $j$'s *TSS*. The matrices $V^j$ $F^j$ and $C_j$ are of dimensions $(m_j \times s)$, $(m_j \times q)$, and $(m_j \times t)$ respectively, where $s$, $q$ and $t$ are the number of annotations used. *BAGEA* models gene expression as a linear combination of SNP genotypes:

$$y_j = X_j b_j + \epsilon_j, \tag{1}$$

where $\epsilon_j$ is an *i.i.d* normal noise vector and $b_j$ is a vector of SNP effects. The effect of SNP $i$ on gene $j$ $b_{ij}$ is modeled as:

$$b_{ij} \sim N((\boldsymbol{\omega}^T \boldsymbol{v}_i^j)(\boldsymbol{\nu}^T \boldsymbol{f}_i^j), \alpha_{ji}^{-1}), \tag{2}$$

Detailed descriptions of the terms are as follows:

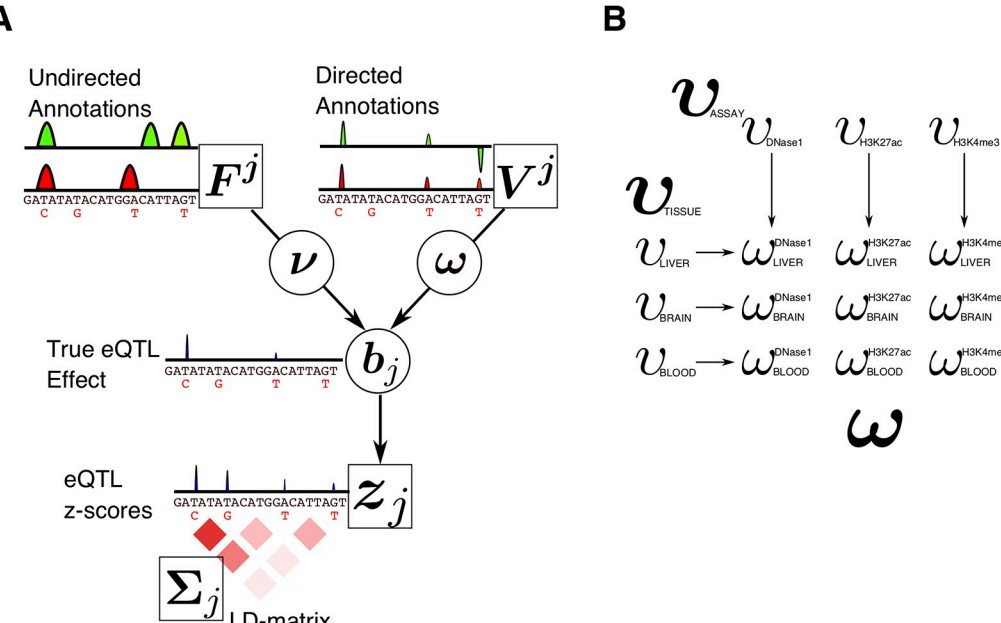

**Fig 1. Illustration of *BAGEA* model components.** (A) The core components of the *BAGEA* model in the summary statistics formulation. Observed variables are in squares while estimated variables are circled. Given are $z_j$, the eQTL z-scores for gene $j$, as well as the LD matrix $\Sigma_j$, defining the correlation between summary statistics. Further, z-scores are influenced by the true eQTL effects $b_j$. These effects in turn depend on directed and undirected annotations, $V^j$ and $F^j$ respectively. While undirected annotations can cover regions of any size, directed annotation have the same size as the genomic variants themselves. The impact of annotations on $b_j$ is estimated from the data via $\omega$ and $v$. (B) An example of the modeling of different priors of elements of $\omega$ using meta-annotations via $v$ variable vectors. We assume that directed annotations are available for nine annotations, which were derived from tissues *Liver*, *Blood* and *Brain* via 3 assay types *DHS*, *H3K27ac* and *H3K4me3*. It is reasonable to assume that for a given eQTL study, particular tissues or cell types are more relevant than others. We model this by introducing a variable $v$ for each tissue (or cell type) that affects the prior distribution of only those elements of $\omega$ that are derived from this tissue, e.g. $v_{Liver}$ only affects elements of $\omega$ tied to experiments performed in liver. We model different priors for various for assay types analogously. Shown is the resulting network of influences of the variable $v_{tissue}$, $v_{assay}$ on $\omega$. (We used the actual group names as indices, while in the main text, elements of $v$'s and $\omega$ are indexed by natural numbers).

- $v_i^j$ encodes the directed annotaton for SNP $i$ in the region of gene $j$. It is the $i$th row of $V^j$. In our applications of *BAGEA*, $V^j$ is previously computed from sequence-based models, where column of $V^j$ represents an epigenetic mark and each row $V^j$ represents a SNP. Each entry in $V^j$ expresses the predicted effect of a genotype change on the epigenetic mark in question.

- $f_i^j$ encodes the undirected annotations for SNP $i$ in the region of gene $j$. It is the $i$th row of $F^j$. Each element in $F^j$ expresses the presence or absence of the annotation at a SNP's location. In our applications of *BAGEA*, $F^j$ is derived from the relative positions of a SNP and gene $j$'s *TSS*, where each column represents a particular region around the *TSS*. For example, if a column $F^j$ encodes a region of 20 kilobases (KB) upstream from the *TSS*, all entries for rows corresponding to SNPs within 20 KB upstream of that *TSS* will be set to 1 and entries for all other rows will be set to 0. By default, the first column of $F^j$ is an intercept column consisting only of ones.

- $\omega$ and $v$ are vectors of lengths $s$ and $q$ respectively that are estimated by *BAGEA*. Specifically, $\omega$ and $v$ are the effects of annotations in $F^j$ and $V^j$ on the SNP effects $b_j$. By default, the effect of the interecept weight $v_1$ is fixed at close to 1 via constraining priors.

- $\alpha_{ij}$ is a scalar estimated by *BAGEA* and models the variance of $b_{ij}$. Allowing different variances for the elements of $b_j$ typically produces sparse estimates for $b_j$ with many elements close to zero as integrating out the Gamma distributed prior $\alpha_{ij}$ yields a t-distribution for the different $b_j$ [18]. Further, $\boldsymbol{\alpha}_j$, the vector of length $m_j$ with $\alpha_{ij}$ its $i$th element, is modeled as dependent on the undirected annotation matrix $C_j$. $C_j$ can potentially be identical to $F^j$ but can model different undirected annotations as well (see Method Details).

Typically, directed annotations are grouped by their cell type or assay type. *BAGEA* can use this grouping structure in order to select groups of annotations that are useful for predicting gene expression (Fig 1B). *BAGEA* selects annotation groups via a modeling strategy that yields sparsity on the annotation group level similar to the group lasso [19]. In *BAGEA*, this grouping strategy is implemented by partitioning annotations into multiple meta-annotations (such as different cell types, assay types etc.). When using this partitioning mechanism, *BAGEA* includes an extra random variable vector $v$ of the same length as the number of elements in the partition structure (e.g. the number of cell types, or the number of assay types) (See Methods as well as Fig 1B for an illustrative example). The $k$th element of $v$, $v_k$, controls the variance of the effect sizes for annotations that fall into partitioning group $k$. Specifically, $v_k$ is proportional to the inverse of the variance of the respective elements of $\boldsymbol{\omega}$. $v_k^{-1}$ is therefore called the *variance modifier* of annotation partition element $k$ (see Methods).

Importantly, the model can be reformulated in terms of the summary statistics $z_j = X_j^T y_j / \sqrt{n}$ and LD matrices $\Sigma_j = X_j^T X_j / n$. The reformulation enables the application of *BAGEA* to studies for which only summary statistics are available, by estimating $\Sigma_j$ from external sources (see Methods).

## Evaluation strategy for model fit

We developed an approach to evaluate the performance of *BAGEA* when fitting directed annotations to genotype and gene expression data. An important feature of *BAGEA* is that its results can be used to predict gene expression for a gene without using any expression data for that gene, but rather using genotypes and genome annotations whose weights are fitted from other genes. We can therefore validate *BAGEA* by training it on gene expression data for one set of genes, and then calculating the extent to which the trained model predicts gene expression for other genes.

We propose a so-called directed predictor $\hat{\boldsymbol{\mu}}_j$, which predicts gene expression for gene $j$ based on knowledge of directed annotations and genotype for gene $j$. Set $\hat{\boldsymbol{\eta}}_j$ as the expected mean shift of $b_j$ due to the annotations. Using the same notation as in Eqs (1) and (7), we have

$$\hat{\eta}_{ij} = E[b_{ij} | v = \hat{v}, \boldsymbol{\omega} = \hat{\boldsymbol{\omega}}] = (\hat{\boldsymbol{\omega}}^T \boldsymbol{v}_i^j)(\hat{\boldsymbol{v}}^T \boldsymbol{f}_i^j), \tag{3}$$

i.e. $\hat{\eta}_{ij}$ is the $i$th element of $\hat{\boldsymbol{\eta}}_j$. the predictor $\hat{\boldsymbol{\mu}}_j$ is then computed by

$$\hat{\boldsymbol{\mu}}_j = E[y_j | v = \hat{v}, \boldsymbol{\omega} = \hat{\boldsymbol{\omega}}] = X_j \hat{\boldsymbol{\eta}}_j. \tag{4}$$

The squared magnitude $S_j = \hat{\boldsymbol{\mu}}_j^T \hat{\boldsymbol{\mu}}_j$ measures how much gene expression variance the model attempts to explain via the predictor $\hat{\boldsymbol{\mu}}_j$. To evaluate the predictor's accuracy and degree of overfitting, we use the *directed mean squared error* $MSE_j^{dir} = (y_j - \hat{\boldsymbol{\mu}}_j)^T (y_j - \hat{\boldsymbol{\mu}}_j) / n$. The evaluation of the predictor is performed on a set of genes independent of the ones used to estimate $\boldsymbol{\omega}$ and $v$.

We can reformulate $MSE_j^{dir}$ in terms of summary statistics $\boldsymbol{z}_j = \boldsymbol{X}_j^T \boldsymbol{y}_j / \sqrt{n}$, LD matrices $\boldsymbol{\Sigma}_j = \boldsymbol{X}_j^T \boldsymbol{X}_j / n$, and $\hat{\boldsymbol{\eta}}_j$:

$$MSE_j^{dir} = 1 - 2\hat{\boldsymbol{\eta}}_j \boldsymbol{z}_j / \sqrt{n} + \hat{\boldsymbol{\eta}}_j^T \boldsymbol{\Sigma}_j \hat{\boldsymbol{\eta}}_j, \tag{5}$$

if we assume that $\boldsymbol{y}^T \boldsymbol{y} = n$. In principle, the reformulation allows us to calculate a predictor's directed mean squared error, even if only summary statistics are available, by approximating $\Sigma_j$ from external sources.

## Directed annotations derived from blood can partially explain *cis*-eQTLs in monocytes

We used *BAGEA* to determine the extent to which annotations can predict gene expression in *CD14* positive monocytes. To this end, we aggregated data from two eQTL studies on expression genetics in *CD14* positive monocytes [20, 21]. For directed annotations, we used predictions of genetic variant effects on epigenetic marks (12 different histone mark assays and *DNase1 Hypersensitivity Site* (DHS) calls with 4 different peak calling strategies) in various blood-derived cell types from the pre-trained *ExPecto* model. *ExPecto* is a deep learning framework that predicts epigenetic marks based on sequence context and performs *in silico* mutagenesis to evaluate the consequences of sequence variants [7]. *ExPecto* yielded 2002 directed annotations of which 253 were from blood related cell types. These are referred to as the *Blood* annotation subset in this paper. We partitioned these directed annotations by cell type and assay type, respectively, and modeled separate prior variance terms for each partition (Fig 1A).

To train *BAGEA*, we used gene expression data from human chromosomes 1 through 15. Only 2410 genes that had a SNP in *cis* showing a signficant association with a p-value lower than $10^{-10}$ were included. To test model fit, we predicted expression for 917 genes on chromosomes 16 through 22 with a top nominal *cis*-eQTL p-value below $10^{-10}$. Specifically, we used the model fit on the training set to derive the estimates $\hat{\boldsymbol{\omega}}$ and $\hat{\boldsymbol{v}}$ (see Eq 7). We then used these estimates to calculate the directed predictors $\hat{\boldsymbol{\mu}}_j$ for genes on the test set (see Eq 4). To assess the predictive power of $\hat{\boldsymbol{\mu}}_j$, we calculated $MSE_j^{dir}$ for every gene in the test set.

We observed that directed genome annotations can partially explain gene expression variance (Fig 2). The average $MSE^{dir}$ across all genes was 99.5%, which was significantly smaller than 100% (as evaluated by bootstrap sampling genes; p-value smaller than $10^{-4}$). $MSE_j^{dir}$ showed a dependence on predictor size $S_j$ (where $S_j = \hat{\boldsymbol{\mu}}_j^T \hat{\boldsymbol{\mu}}_j$), such that for the top quartile of genes when ranked by $S_j$, the directed component was estimated to predict 1% to 3% of expression variance (Fig 2A). For each gene, the variance explained is bounded by the additive genetic variance component in *cis* which is typically much lower than 100%. We estimated the variance of expression explained for each gene in *cis* in an unbiased way via Haseman-Elston (HE) regression [22]. This approach suggested that around 6.6% of the total genetic variance in *cis* was explained by the externally fitted directed component $\hat{\boldsymbol{\mu}}_j$ for genes in the top quartile w.r.t $S_j$ (Fig 2B). Across all strong cis-eQTLs, we estimated that the directed component explained 2.5% of total additive genetic variance in cis. We further tested the impact of the distance modifier by constraining all elements of $\hat{\boldsymbol{v}}$ (except the intercept element) to zero, showing that the modeling the distance modifier leads to higher predictive power (S1 Fig).

These results show that *BAGEA* can be used to model how sequence changes affect gene expression. Note that this evaluation metric relies on global parameter (i.e. $\boldsymbol{\omega}$, $\boldsymbol{v}$) estimates only. This allows to form predictors for a gene's expression even if the gene was not included in the training set, but has lower predictive power than approaches that use genewise local parameter estimates (i.e. $\boldsymbol{b}_j$). These approaches can predict expression potentially in an

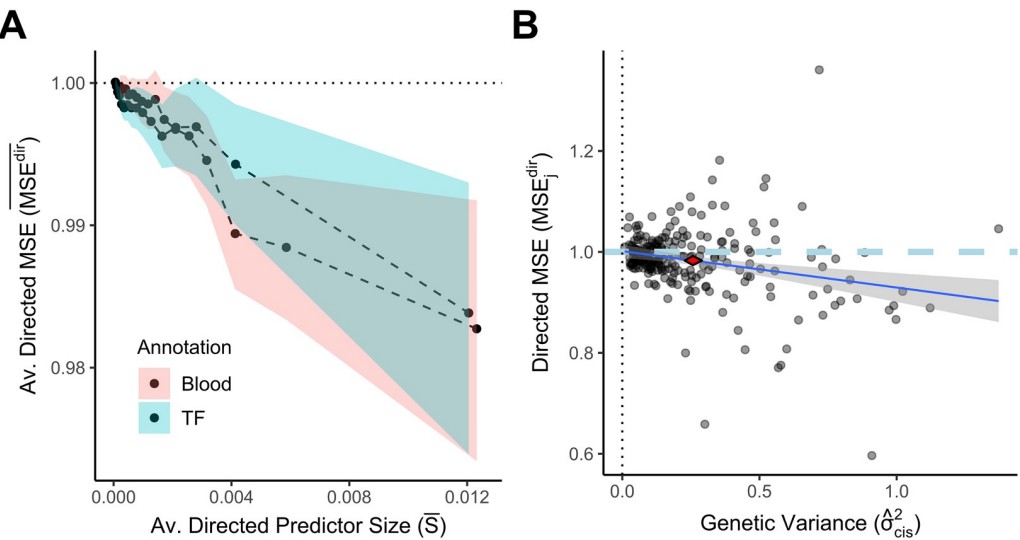

**Fig 2. Gene expression variance can be partially explained by directed genome annotations.** The *BAGEA* model was fitted on genes in the training set (all genes on chromosomes 1 through 15) using monocyte eQTL data on genes with a top nominal p-value below $10^{-10}$, and with ExPecto-derived directed annotations. ExPecto includes 2002 annotations in total, of which one of two subsets were used: 253 annotations derived from histone and DHS assays in a blood related cell types (*Blood*), or, alternatively, 690 annotations derived from TF ChIP-Seq (*TF*). For each gene $j$ in the test set (all genes on chromosomes 16 through 22 with a top nominal p-value below $10^{-10}$), we calculated the directed predictor of expression $\hat{\boldsymbol{\mu}}_j$. As a measure of a predictor's size, we use its squared magnitude $S_j = \hat{\boldsymbol{\mu}}_j^T \hat{\boldsymbol{\mu}}_j$. To evaluate the predictor's performance, we calculated $MSE_j^{dir}$, the mean squared error (*MSE*) when predicting gene expression $\boldsymbol{y}_j$ from $\hat{\boldsymbol{\mu}}_j$. To estimate what the smallest attainable $MSE_j^{dir}$ would be, we estimated $\sigma_{g_{cis}}^2$, the additive genetic variance in *cis* via Haseman-Elston regression per gene. (A) The relationship between the MSE of the predictor and its squared magnitude. We sorted results by predictor Size $S_j$ and averaged $MSE_j^{dir}$ within a sliding window containing 25% of genes and step size of 5% of data. **Averaged Directed Predictor Size** $\bar{S}$: The mean value of $S_j$ per window on the horizontal axis; **Averaged Directed MSE** ($\overline{MSE^{dir}}$): The averaged $MSE_j^{dir}$ of genes falling into the window on the vertical axis. The 95% confidence interval for each window was derived by bootstrapping. Most variance is explained by genes in the top quartile when ranked by $S_j$. (B) The relationship between $MSE_j^{dir}$ and $\sigma_{g_{cis}}^2$ for genes in the top quartile when ranked by $S_j$. **Genetic Variance** ($\sigma_{g_{cis}}^2$): The estimated additive genetic variance in *cis* on the horizontal axis. **Directed MSE** ($MSE_j^{dir}$) on the vertical axis. 95% confidence intervals for the mean of both the $MSE^{dir}$ and $\sigma_{g_{cis}}^2$ are represented as the corners of the red diamond (i.e. the confidence interval for the average $MSE^{dir}$ is given by the upper and lower corner, whereas the confidence interval for the average $\sigma_{g_{cis}}^2$ is given by the right and left corner respectively). A linear regression is plotted as the blue line, with 95% confidence interval shown in grey.

out-of-sample fashion, but only for genes in the training set. To illustrate this, we fit *BAGEA* on a subset of the monocyte samples (134 samples from the *Fairfax et al.* study) and extracted the local parameter estimates $\hat{\boldsymbol{b}}_j$ [20]. We then used these estimates to predict gene expression in the other available monocyte samples [21] [20]. As expected, a substantial fraction of the genetic variance in *cis* could be predicted using the local parameter estimates (S2 Fig). Further, using *BAEGA* in an annotation uninformed manner lowered the variance explained.

## Joint modeling of cis-eQTLs and directed annotations highlights biologically relevant epigenetic marks

We next evaluated if *BAGEA* can effectively be used to discover which annotations, or groups of annotations, are most predictive of gene expression. We grouped the directed annotations by cell type and assay type, and for each set of annotation groups, we modeled separate prior variance modifiers $\boldsymbol{v}^{-1}$ (Fig 1B). For each annotation group $k$ we measured its contribution to gene expression as its estimated variance modifier $v_k^{-1}$ (See Model Overview). For the

monocyte data, *BAGEA* estimated the largest variance modifiers for annotations from *DHS* as well as *H3K27ac* and *H3K4me3* assays (Fig 3A). This observation is consistent with results from a previous method, using undirected annotations, suggesting that SNPs with an effect on gene expression are enriched in open chromatin (*DHS*), activated enhancers and promoters (*H3K27Ac*, *H3K4me3*) [11]. Across cell type annotations, *BAGEA* estimated the largest variance modifiers for annotations from two blood cell types that were both *CD14* positive (Fig 3B). This observation matches our expectations because the cells in the underlying expression data were derived from *CD14* positive cells [20, 21]. Across all tested pairs of assays and cell types, *BAGEA* estimated the largest positive effect sizes for annotations from *DHS*, *H3K27ac*, *H3K4me3* assays in *CD14* positive cells (Fig 3C). Additionally, we saw a negative effect size for *DHS* assayed in *CD3* positive cells, albeit with lower absolute effect size than *CD14* positive cells. One explanation could be that *DHS* that occur in *CD14* but not *CD3* positive cells have larger predictive value than *DHS* that occur in both cell types.

It is well known that eQTLs occur more likely and increase in effect size closer to the *TSS*. This suggests that the effects of directed annotations might also be bigger for SNPs close to the *TSS* than for SNPs that are distal. *BAGEA* models SNP distance dependence of directed annotation effects by weighting the directed annotation effect term $\boldsymbol{V^j\omega}$ across SNPs, with a distance modifier $\boldsymbol{F^j\nu}$ (see Model Overview). We next tested whether *BAGEA* estimated directed annotation effect sizes to be dependent on a SNP's distance to the *TSS*. We examined the value of a SNP's estimated distance modifier $\boldsymbol{F^j\hat{\nu}}$ against its position relative to the *TSS*. We observed a characteristic peak around the *TSS* (Fig 3D), suggesting that *BAGEA* can indeed produce a similar pattern of distance dependence for the effect sizes derived from directed annotations as for the eQTL effect sizes themselves.

We repeated this analysis with a different set of directed annotations, namely 690 *ExPecto* annotations derived from transcription factor (*TF*) ChIP-Seq in any cell type. We estimated the *TF* annotation subset to be similarly predictive of gene expression as the *Blood* annotation subset (Fig 2A). When looking at the estimates of $\boldsymbol{\omega}$, *MYC* assayed in the cell line *NB4* had the largest effect size among all tested 690 annotations (S3 Fig). Additionally, *SPI1*, *MAX CTCF* had effects larger than 10% of the maximal effect size. For *SPI1* and *CTCF*, effects from multiple cell lines reached this threshold with consistent effect size directions.

*NB4* is a promyelocytic leukemia cell line that can be differentiated into neutrophils or monocytes [23]. *NB4* is therefore expected to have similar expression genetics as *CD14* positive monocytes, and, given that no *TF* ChIP-Seq experiment was performed in monocyte cell lines directly, the large $\boldsymbol{\omega}$ values for *NB4* data are consistent with our expectations. However, interpretations of cell type selection for the TF subset are complicated by the fact that the underlying *TF* ChIP-seq experiments did not sample each *TF* comprehensively across all cell lines which might lead to biases. When checking expression of the highlighted *TFs* in monocytes via *Protein Atlas*, we found all were *TFs* classified as expressed but not elevated in monocytes [24]. Effect size directions of *CTCF* and *MAX* were negative, which naively interpreted would suggest that their binding have a suppressive effect on gene expression. *CTCF* can act as repressor, activator and insulator [25]. Our data suggests that, globally and in the studied context, repressive effects outweigh. *MAX* and *MYC* are part of a family of *TFs* that form heterodimers [26]. The *MYC/MAX* dimer is usually regarded as an activator, which might seem to be at odds with our results as the *MAX* annotation had a negative effect size. However, another important family member *MXD1* (also known as *MAD*) was not assayed. The *MAD/MAX* heterodimer is thought to act as a repressor. A positive effect size for *MYC* and a smaller negative effect size for *MAX* could just imply that *MAX* binding in absence of concomitant *MYC* binding has a negative effect on expression because it tracks with *MAD/MAX* heterodimer binding.

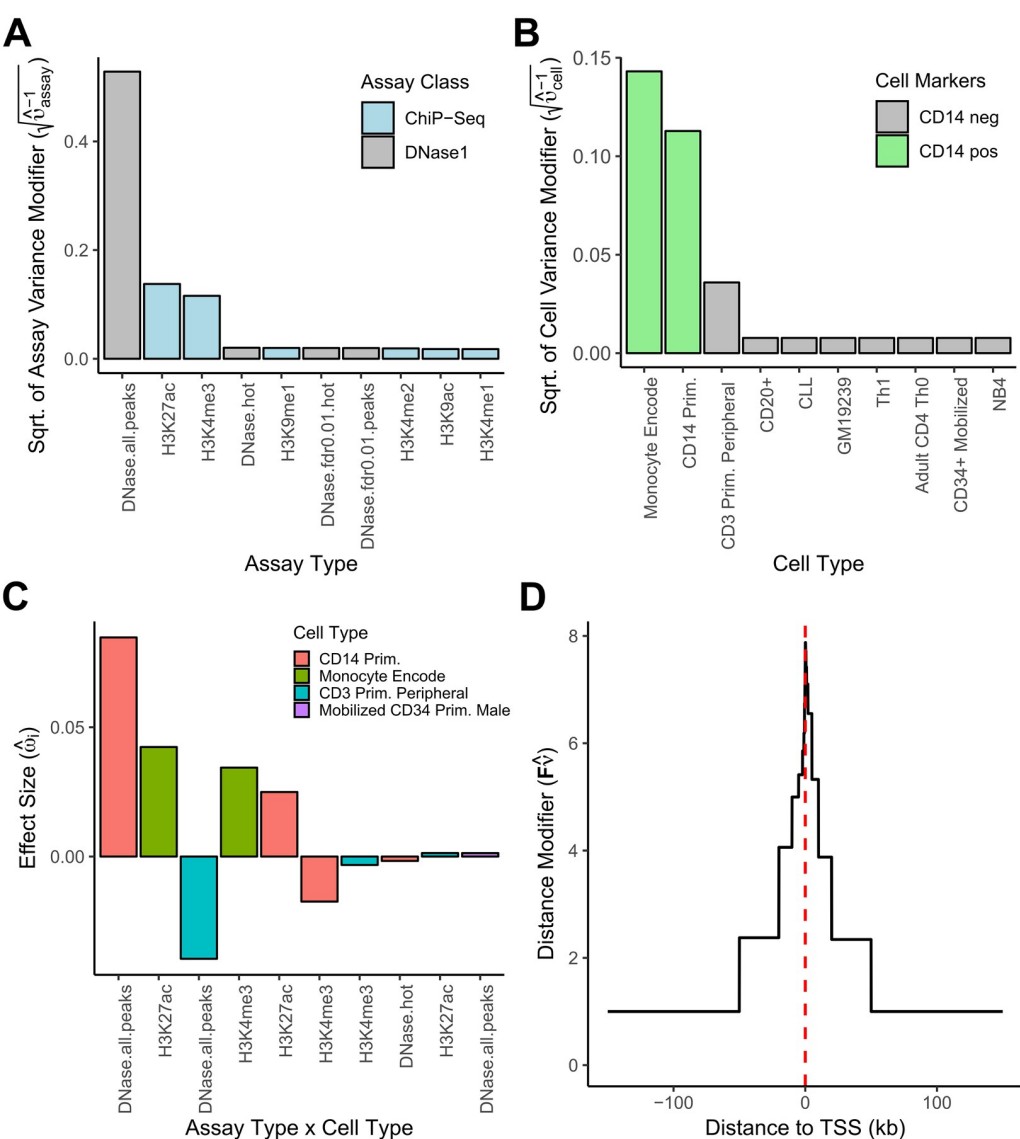

**Fig 3. *BAGEA*, fitted on monocyte eQTL data, selects relevant epigenetic marks and increases directional effect sizes for SNPs close to a TSS.** Parameter estimates when applying *BAGEA* to monocyte eQTL data using as directed annotations histone and DHS *ExPecto* predictions derived from blood-related cell types (i.e. *Blood* from Fig 2). (A) For each chromatin assay type, *BAGEA* models an **assay variance modifier** $\hat{v}_{assay}^{-1}$ that captures the extent to which that assay type is predictive of gene expression. Shown are the square roots for the assay types with the ten highest variance modifiers (from 17 assay types total). In the *BAGEA* model, *DHS*, *H3K27Ac* and *H3K4me3* assays have the largest modifiers. (B) For each cell type, *BAGEA* models a **cell type variance modifier** $\hat{v}_{cell}^{-1}$, similar to the assay variance modifier in panel A. Shown are the square roots for the cell types with the ten highest variance modifiers (out of 61 cell types). In the *BAGEA* model, *CD14* positive cells have the largest modifiers. (C) *BAGEA* reveals experiments underlying the directed annotations that were most predictive of gene expression. **Assay Type x Cell Type**: Each experiment is a particular assay type performed in a particular cell type. **Effect Size** ($\hat{\omega}_i$, for experiment i): The *BAGEA*-estimated effect on gene expression. Shown are the ten largest directed annotation effect sizes. In the *BAGEA* model, the experiments using *DHS*, *H3k27Ac* and *H3Kme4* with *CD14* positive cells have the largest effect sizes. We also see that most of the 253 annotations are estimated to have a close to zero effect. (D) Shown is the estimated **distance modifier** of the directed component, $\mathbf{F}\hat{\mathbf{v}}$. We see a characteristic peak around the *TSS*, implying that the directed annotations are upweighted close to the *TSS*.

Developmentally, during monocyte/macrophage maturation, a switch from high levels of *MYC/MAX* to *MAD/MAX* is observed [27]. A further highlighted transcription factor is *SPI1* which is regarded as one of the most important transcription factor in monocyte and macrophage development [28].

## Modeling directional components is robust to the use of summary statistics

In many cases it is not feasible to compute LD for the population from which the summary statistics were derived (i.e., the study population), and LD has to be derived from other sources (i.e., external genotypes) [29, 30]. The use of external genotypes allows publicly available summary statistics to be analyzed without access to restricted individual level genotype data [9]. However, LD computed on external genotypes can only approximate LD patterns of the study population. We therefore need to test the accuracy of methods when using external genotypes.

We evaluated if directed annotation effects were robust to the genetic source of LD information. We used 1000 Genomes data to compute LD [31]. We re-fit the *BAGEA* model to the monocyte data with the *Blood* annotation subset, using LD matrices derived from European 1000 Genomes data. We then compared $\omega$ estimates when using LD from 1000 Genomes to $\omega$ estimates when using LD from the monocyte data itself, for every annotation in the monocyte Blood data. We observed that the two approaches produced similar effect sizes with a linear regression $R^2$ of 97.5% and regression slope of 0.96 (S4A Fig). This suggests that directed annotation effect estimates are robust to the source of LD information. We then explored if the source of LD information affected our estimates of directed mean squared error ($MSE^{dir}$). To this end, we estimated $MSE^{dir}$ on chromosomes 16 through 22 from summary statistics and external LD matrices derived from 1000 Genomes alone, and then compared these $MSE^{dir}$ values to the original $MSE^{dir}$ values computed with LD derived from monocyte data. We ensured that the same SNPs were included, by removing SNPs with low minor allele frequency (MAF) in either of the sets. We observed that the two sources of LD produced $MSE^{dir}$ values that agree with each other, with a linear regression $R^2$ of 99.9% and regression slope of 1.002 (S4B Fig).

## Exploring *BAGEA*'s ability to identify causal marks through simulation

To explore *BAGEA*'s ability to select the causal annotations among the set of annotations, we used simulation (see S1 Appendix). Naturally, this depends on the correlation structure of the directed annotations, as highly correlated annotations will make it difficult to isolate the causal one. We therefore used empirically observed directed annotations in our simulation experiment. We assumed a model were the truly causal annotations were sparse (with the non-zero effects varying from 3 to 12). We tested cases where the non-zero effects clustered in terms of meta-annotations (e.g. clustered in terms of cell types and assay type) (*structured*) and cases where non-zero effects did not cluster (*unstructured*). While fitting *BAGEA* we also used two parameter settings, either making use of the meta-annotations available for cell type and assay type, (*group-lasso*) or ignoring the meta-annotation information and letting each $\omega_i$ parameter be controlled by a separate $v_i$ parameter (*lasso*). While we saw high recovery of the causal annotations for lower number of causal variables, precision and recall tended to drop as more but individually smaller non-zero effects were added (S5 Fig). Drop-off was fastest when pairing *unstructured* data generation, with the *group-lasso* fitting procedure, presumably, because this parameter setting tried to enforce a structure that was not actually present. Conversely, *structured* data generation paired with *group-lasso* fitting procedure showed the highest performance of all settings. When evaluating the gene expression prediction power of the model fits, we saw that in all cases the results were close to optimal, suggesting that even in higher

complexity settings when incorrect variables get picked, the chosen variables are highly correlated with the correct ones (S6 Fig).

## Building predictive expression models from GTEx summary statistics

Having established that *BAGEA* performs well when using summary statistics, we next determined if *BAGEA* can identify relevant directed annotations for empirical data for which summary statistics are available but genotypes are not. Specifically, we fit *BAGEA* on summary statistics for eQTL studies of 13 tissues produced by the GTEx consortium with a sample size of at least 300 for each study [9]. We additionally supplemented this set with results for Lymphoblastoid cell lines (LCL) derived from a meta-analysis of GTEx and GEAUVADIS [17]. Because GTEx gathered eQTLs in complex tissues and sampled fewer individuals than were sampled in the monocyte studies, we expected lower power to produce robust parameter estimates. We therefore used different parameter values than in our monocyte analysis, including genes with top nominal *cis*-eQTL p-value lower than $10^{-7}$. We fitted models either using *ExPecto* derived annotation for all 1187 histone or DHS annotations derived from Roadmap consortium data or the derived annotations for non-histone ChIP-seq data from ENCODE [32, 33]. When using Roadmap annotations we used *BAGEA* in the *group-lasso* setting, whereas for ENCODE annotations we used the *lasso* setting, the rationale being, that the Roadmap consortium performed most assays for a given cell type, whereas ENCODE ChIP-seq was less complete, i.e. many TFs were assayed in only few cell lines, leading to potentially strong biases.

We again split genes into training and test set, fitting *BAGEA* on the training set and building directed expression predictors $\hat{\boldsymbol{\mu}}_j$ for all genes in the test set. We observed that the average $MSE^{dir}$ per dataset was variable across GTEx datasets ranging from 100% to below 98.5% (Fig 4A). When looking at only the highest quartile of genes in terms of squared magnitude $S_j$, we saw the lowest average $MSE^{dir}$ go to approximately 0.95 (Fig 4B). Furthermore the gains in average $MSE^{dir}$ were in line with squared magnitude $S_j$ values, suggesting that *BAGEA* does not substantially overfit. We saw that the ENCODE *TF* annotation set tended to outperform the histone and DNase1 *Roadmap* set and that in the Roadmap group lasso setting, *BAGEA* would not always select any annotations, potentially due to poor overlap between annotation and GTEx cell types and stringent regularization.

We next compared the predictive power achieved by *BAGEA* on the GTEx data to results derived via ExPecto directly. To predict expression from genomic variants, the authors of ExPecto propose a strategy, where results from two statistical models are combined. The first model is a deep neural network that predicts the impact of genomic sequence variants on chromatin marks (the results of which are also used by *BAGEA* in this analysis). The second model is a $l_2$-boosting model that predicts gene expression from (spatially transformed) chromatin marks directly. as part of the ExPecto release, results of the second model were already publicly available for 13 relevant GTEx datasets [4]. Combining these results with the directed annotations and the z-scores from GTEx, allowed us to estimate the scalar product between the gene epression vector $\boldsymbol{y}_j$ and the corresponding directed predictor $\hat{\boldsymbol{\mu}}_j$ (see S1 Appendix). This allowed us to compare model quality in terms of the fraction of genes with a scalar product larger than zero. When comparing results from *BAGEA* (using TF annotation subset and lasso setting) and ExPecto (using all annotations) in terms of this metric, we saw that, while performance was comparable across all genes, BAEGA substantially outperformed ExPecto for genes in the highest quantiles in terms of effect size (S7 Fig).

We further compared *BAGEA* to *Torus*, a tool which allows to model SNP effect priors in terms of undirected annotations [14, 15]. We therefore made our annotations undirected by

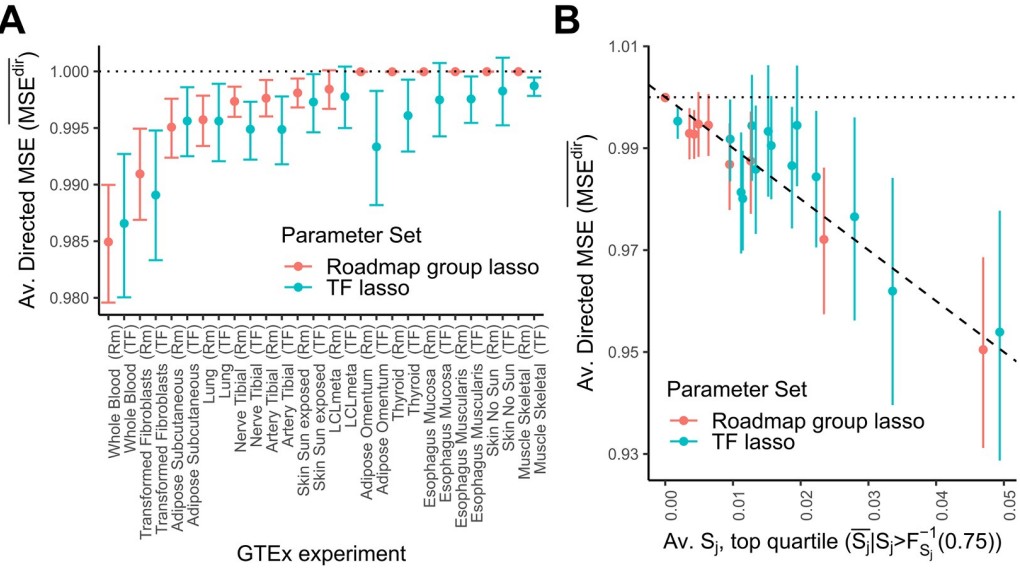

**Fig 4. Directed annotations partially explain gene expression variance in GTEx.** The *BAGEA* model was fit using various GTEx eQTL data (supplemented with GEAUVADIS eQTL data) and with ExPecto-derived directed annotations on genes in the trainig set (chr1,..,chr15) with a top nominal p-value$<10^{-7}$. ExPecto includes 2002 total annotations, of which either 1187 histone and DHS annotations from Roadmap (*Roadmap*) or 690 non-histone ChIP-Seq from ENCODE (*TF*) were used. For the *Roadmap* annotation set we enforced structure on the priors of $\boldsymbol{\omega}$ by using the meta-annotations available for cell type and assay type, (*group-lasso*), while for the (*TF* annotation set, each $\omega_i$ parameter be controlled by its individual $v_i$ parameter (*lasso*). For each gene $j$ in the test set (chr16,..,chr22 and top nominal p-value$<10^{-7}$), we calculated an approximate version of $S_j$, the squared magnitude of the directed predictor $\hat{\boldsymbol{\mu}}_j$, where the approximation uses external LD information. Further, we calculated an approximate version of $MSE_j^{dir}$, the mean squared error (*MSE*) when predicting gene expression $\boldsymbol{y}_j$ from $\hat{\boldsymbol{\mu}}_j$. (A) Displayed is the average (approximated) $MSE_j^{dir}$ across all genes for each GTEx experiment, and annotation subset. 95% Confidence intervals are computed by bootstrap sampling. (B) For each GTEx experiment and annotation subset, we sorted results by predictor size $S_j$ and and averaged $MSE_j^{dir}$ within the top quartile. Displayed is the relationship between the MSE of the predictor and its mean squared magnitude $S_j$. **Averaged** $S_j$, **top quartile** $\bar{S}_j|S_j > F_{S_j}^{-1}(0.75)$: The mean value of the directed predictor size $S_j$ in the top quartile on the horizontal axis; **Averaged Directed MSE** ($\overline{MSE^{dir}}$): The averaged $MSE_j^{dir}$ of genes falling into the top quartile in terms of $S_j$ on the vertical axis. The 95% confidence interval for each window was derived by bootstrap sampling. We see that the average squared magnitude $S_j$ is of similar size as the gains in directed *MSE* suggesting that the *BAGEA* does not substantially overfit.

taking absolute values and adding the undirected annotations used in *BAGEA* to generate annotations for *Torus* (see S1 Appendix). We fit data from 13 GTEx experiment on the TF annotation subset filtering SNPs and genes as for *BAGEA*. Varying amounts of $l_2$ regularization were applied and an additional *overfit* strategy was added to have an upper bound on results achieved with an optimal regularization strategy (see S1 Appendix). As an evaluation strategy we recorded for each gene in the test set the SNP with the highest prior. As an evaluation metric, we used the fraction of genes for which that SNP had an absolute z-score close to the largest absolute z-score for that gene. We saw that, while *BAGEA* had a slightly higher value than even the *overfit* strategy, the increases were modest (S8 Fig). As *Torus* was run in ridge mode, few effects were very close to zero. *BAGEA* in its default parameter resembles lasso, in that it only selects a limited number of effect sizes substantially different from zero. To compare the variables selected, we split the distribution torus effect sizes estimates into 3 groups based on whether *BAGEA* estimated them as (close to) zero, negative or positive. We saw that the distribution of torus estimated effect sizes was substantially shifted for the non-zero *BAGEA* effect groups compared to the zero *BAGEA* effect group (S9 Fig).

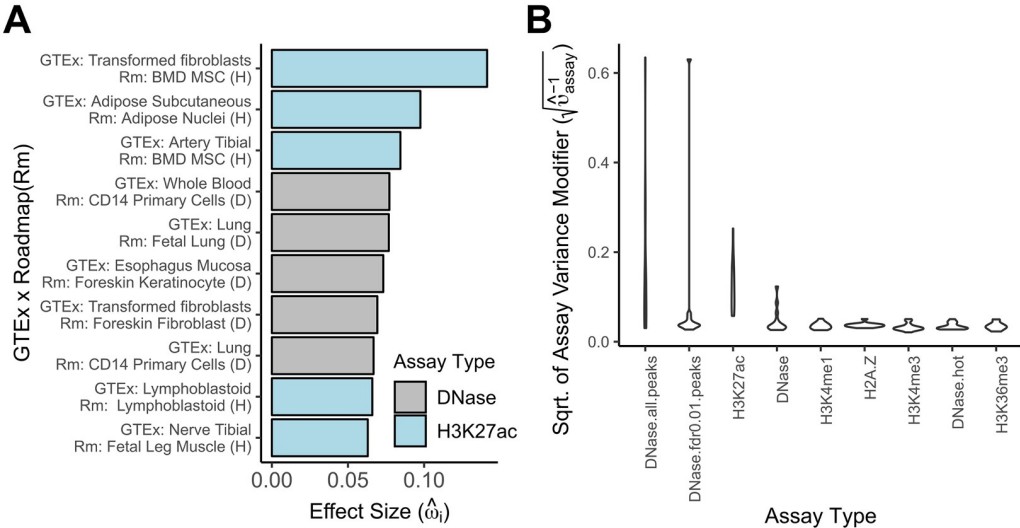

**Fig 5. Model fit for GTEx summary statistics selects directional annotations mainly from biologically consistent cell types.** Shown here are various parameter estimates from fitting 13 different GTEx eQTL summary statistics data (supplemented with GEAUVADIS eQTL data) using histone and DHS *ExPecto* predictions derived from Roadmap (1187 annotations). (A) *BAGEA* reveals the experiments underlying the directed annotations that are most predictive of gene expression. **GTEx x Roadmap(Rm)**: Each GTEx eQTL dataset highlights particular Roadmap annotations. Shown here are the 10 largest positive effect sizes across all eQTL and annotation pairings. **Effect Size**: The estimate of $\hat{\omega}_i$ for experiment *i*. (B) For each chromatin assay type, *BAGEA* models an assay variance modifier $\hat{v}_{assay}^{-1}$ that expresses the extent to which that assay type is predictive of gene expression. Shown here is the distribution of the square roots of the assay variance modifier for any given assay type across all 13 GTEx eQTL datasets. Results are sorted by the maximal value achieved for each assay type and only the 10 highest scoring assay types are shown. We see that *DNase.all.peaks H3K27ac* annotations dominate. The *DNase.fdr0.01.peaks* was prioritized in Lung tissue, which had the lowest value for *DNase.all. peaks* among all experiments. The highest value in *DNase.all.peaks* was achieved in Fibroblast, an experiment that also showed low average $MSE^{dir}$.

## GTEx expression models make use of biologically relevant annotations

To mitigate the impact of limited power during variable selection, we additionally fit models without splitting chromosomes into test and validation sets. Exploring the results from the Roadmap annotation subset first, we saw that the distribution of effect sizes of the directional annotations revealed a bias towards positive values (S10 Fig). Focusing on the largest positive effect sizes(top ten or $\hat{\omega}_i > 0.06$), we saw many biologically consistent pairings between the tissue assayed by GTEx via eQTL and the tissue assayed by Roadmap for epigenetic marks (Fig 5A). While some of the pairings are obvious from the annotation names themselves (such as correct pairings for lymphoblastoid cells, lung and adipose tissues) others were suprising yet on closer inspection, turned out to be consistent with the biological literature. For instance bone marrow derived mesenchymal stem cells (*BMD MSC*) are paired with fibroblast. A recent study found no functional differences between the two cell types leading the authors to support a longstanding opinion in the field that these two cell types should be classified as the same [34, 35]. The pairing between Esophagus Mucosa and keratinocytes can be explained by the fact that the Esophagus Mucosa is mainly composed of squamous cells, i.e. keratinocytes [36, 37]. The pairing between tibial artery and *BMD MSC* can be explained by the fact that fibro-blasts are the main component of vascular adventitia [38]. Our model also paired tibial nerve and muscle, which seems physiologically the least biologically consistent among the ten pair-ings. When looking at the largest negative values, we saw some of the same tissue pairings repeated, with only one pairing with effect size $\hat{\omega}_i$ smaller than -0.06 for the pairing between

fibroblasts and *BMD MSC*) (S11 Fig). When looking at the variance modifier estimates for the different assay types, we saw that *DHS* and *H3K27ac* epigenetic marks were ranked consistently highly (Fig 5B). Interestingly, among various annotations derived from the same *DHS* experiments, some performed consistently better than others: *DNase1* peak call annotations outperformed *DNase1* hotspots calls. These two annotation types make use of the same underlying DNAse-seq data, but use different data processing pipelines [39]. Hotspots calls have variable length and are typically wider than peak calls which have a fixed length of 150 bp. *BAGEA* makes a clear choice which should be preferred to model sequence impacts on gene expression. Turning our attention to the results for the non-histone ChIP-Seq ENCODE annotation subset, we saw for most GTEx experiments, that the largest effect size was associated with a *Pol2* assay (S12 Fig). As all annotations are fit together, it is possible, that large *Pol2* effects could obscure transcription factor action as Pol2 binding could be a result of the binding of other factors. We therefore removed all *Pol2* assay annotations, and refit the model again. We then counted the number GTEx experiment for which a given TF showed either a substantial positive or negative effect (S13A Fig). When looking for TFs That showed negative effect sizes in at least half of the assayed experiments, we found *EZH2*, *SIN3A* and *ZEB1*, which all have substantial literature support for being transcriptional repressors [40–42]. TFs that showed substantial positive effects in at least half of the experiments, we saw *PHF8* and *ELF1*. *PHF8* is known to demethylate *H3K9me2*, a mark strongly associated with transcriptional repression [43]. The fact that this assay shows significant positive effects in most analysed GTEx experiments, might highlight an underappreciated importance of this mode of transcriptional control. *ELF1* has been cited in the literature as having both activator and repressor properties [44, 45]. Our results suggest that activator properties outweigh. To systematically evaluate whether our results are in line with the literature, we compared them to the most recent human version of TRRUST, literature database of regulatory interactions between TFs and their targets [46]. Interaction in TRRUST are annotated as repressive or activating in nature. We derived a TFs repressor activity score based on the fraction of annotated interactions defined as repressive. We derived a second TF repressor actvity score as the number of positive effects minus the number of negative effects (S13B Fig). We saw a strong correlation between these scores (p-value below 0.0001, $R^2 = 0.43$). When removing the 3 bona-fide repressors, results remained significant at the 0.05, level albeit less so (one-tail p-value below 0.015, $R^2 = 0.15$). Additionally, we saw that while distance modifier did vary between fits, the characteristic peak around the *TSS* was replicated in all cases (S14 Fig).

## Discussion

Here we introduced a new method, named *Bayesian Annotation Guided eQTL Analysis* (*BAGEA*). *BAGEA* integrates directed and undirected genome annotations in a multivariate fashion with eQTL data in a variational Bayesian framework to build predictive models of gene expression. We applied this method to eQTL results from *CD14* positive monocytes as follows: First, we derived directed annotations by predicting functional impacts on epigenetic marks for all common SNPs using the pre-trained *ExPecto* deep neural net [7]. Second, from these *ExPecto* results, we extracted two annotation subsets of particular interest: histone ChIP-Seq and DHS in blood-derived cell types (the *Blood* annotation subset), and TF ChIP-Seq in any cell type (the *TF* annotation subset). We then ran *BAGEA* on both annotation subsets separately, while allowing the effect of the directed annotations to depend on the distance to the *TSS*. We tested whether the model had explanatory power with a training and test protocol (i.e. explanatory power was estimated on genes that were excluded from training). We saw that the directed component $\mu$ of the model explained part of the gene expression variance in a

statistically significant manner (Fig 2A). For genes with a strong cis-eQTL (p-value$<10^{-10}$) and in the top quartile for $\boldsymbol{\mu}^T\boldsymbol{\mu}$, we estimated that the *Blood* derived directed component explained 6.6% of total additive genetic variance in *cis* (Fig 2B). Importantly, *BAGEA* prioritized annotations that cohere with widely accepted biological knowledge and are supported by existing literature (Fig 3).

Next, we investigated to which extent the model fit was affected when the LD information was approximated via reference genomes. We observed agreement between the results in terms of the directed component, suggesting that the use of eQTL summary statistics together with external LD data is justified (S4 Fig). We next used simulation to investigate, whether *BAGEA* is able to reliably detect causal annotations and found that to be the case if the generating model was sufficiently sparse S5 Fig. We then used *BAGEA* to analyze eQTL summary statistics results from GTEx. To accommodate the wide range of tissues explored in GTEx, we expanded the number of directed annotations used in the fitting process to over a thousand. While for some tissues, the analysis strategy was underpowered to derive a predictive model of gene expression from directed annotations, others had a significant fraction of gene expression explained by directed annotations (Fig 4). We compared results from *BAGEA* to both ExPecto for the ability predicting gene expression from annotations alone and *Torus* for predicting causal SNPs from annotations alone. In terms of predicting gene expression *BAGEA* showed favourable performance for genes with large effect sizes in terms of $S_j$. While *BAGEA* did show improvement over *Torus* w.r.t. prediction of causal SNPs, the improvement was marginal. One plausible explanation is that distance to TSS is already a very strong predictor of causal SNP location. Many of the directed annotations *BAGEA* selected were derived from tissues that were biologically related to the original tissue of the eQTL studies (Fig 5A). Additionally, we observed that *DNAse1* and *H3K27ac* epigenetic marks were selected across many different eQTL studies (Fig 5B). Furthermore, we used the results to classify TFs *de novo* into transcriptional activators and repressors, showcasing an application that relies crucially on directed annotations S3 Fig.

*BAGEA* belongs to a class of models that allow the prior probability distribution of a SNP's effect size to vary based on the genome annotations with which it overlaps [11–13]. These prior models explored the impact of undirected annotations. While *BAGEA* can model undirected annotations, the main novelty comes from the concomitant modeling of directed and undirected annotations as well as interactions thereof. Using directed annotations to explain natural variation in phenotypes was also recently proposed by both *Zou et al.* and *Reshef et al.*, albeit with different modeling philosophies [7, 10]. *Zou et al.* use a model that predicts expression from chromatin patterns directly. This has the advantage that genotype data is not needed. However, this method does not model the causal impact of epigenetic marks on expression levels but rather correlations between them. This modeling approach therefore assumes *a priori* that causality flows from epigenetic marks to gene expression. However, recent integrative analysis modeling causality between expression and chromatin marks suggest that this is not always the case as expression can itself reorganize proximal epigenetic patterns [47].

*Reshef et al.*'s LD profile regression method has more similarities to *BAGEA* as it can also be used to analyze directed annotations and eQTL summary statistics. However, the method is geared towards multiple hypothesis testing rather than high predictive accuracy. Compared to *BAGEA*, the fitted model is simpler allowing for fast analysis of large collections of data. The increased speed comes at the cost of not being able to model certain features like interactions of directed and undirected annotations (such as distance to *TSS*). *BAGEA* uses a modeling approach that has both prediction and interpretability in mind. It allows for more complex model features while still being useful for revealing relevant biology. Indeed, when using

*BAGEA* on various eQTL datasets, *BAGEA* highlighted many relevant cell types. Further, allowing the directed component to depend on the distance to the *TSS* improved the model fit S1 Fig.

There are at least two drawbacks to *BAGEA*'s model complexity. First, there is a substantial computational cost to fit the model. To mitigate this issue, we used various computational strategies such as fast matrix inversion of approximated LD matrices and parallelization (see S15 Fig for an overview of the duration of various analyses). Second, variational model fitting approach does not provide confidence intervals. While it does provide credibility intervals, the approximative nature of mean field variational inference makes these credibility intervals often unreliable [48]. In our analysis, we opted for evaluating statistical significance of the model results by using a training and test protocol.

Future research could investigate whether using a different variational approximation rather than the mean field approximation provides better estimates of the true credibility intervals. An avenue also not explored here, is to learn across datasets by adjusting the priors. The bayesian nature of our framework offers a simple iterative strategy here: After fitting various datasets once, adjust the priors according to the results seen across the different datasets and refit. Whether this strategy can substantially improve predictive power remains to be seen. Further, while the method can be extended to predict expression effects of rare variants, we focused here on prediction of relatively common SNPs. With the cost of whole genome sequencing (WGS) dropping, WGS eQTL studies suited for this purpose should become widely available.

We estimated the extent to which epigenetic marks are able to predict the genetic component of gene expression in *cis*. Our results show that while the current generation of directed annotations can partially explain the genetic *cis* component of gene expression, most of the genetic *cis* component remains unexplained, indicating that there is still room for improvement. Future gains in this space will likely come from both improved directed annotations as well as improved modeling.

## Methods

### Model details

We assume individual level genotype and expression data for $n$ individuals. For gene $j$, we model its $n \times 1$ expression vector $\boldsymbol{y}_j$ as

$$\boldsymbol{y}_j = \boldsymbol{X}_j \boldsymbol{b}_j + \boldsymbol{\epsilon}_j, \tag{6}$$

where $\boldsymbol{X}_j$ is the $n \times m_j$ genotype matrix for the $m_j$ SNPs surrounding gene *j*'s *TSS*. $\boldsymbol{b}_j$ is the $m_j \times 1$ vector of SNP effect sizes and $\boldsymbol{\epsilon}_j$ the expression noise unexplained by the genotype.

$$\boldsymbol{\epsilon}_j \sim N_n(\boldsymbol{0}, (\lambda_j)^{-1} \boldsymbol{I}_n).$$

The noise term precision $\lambda_j$ is modeled in a hierarchical fashion:

$$\lambda_j \sim \Gamma(\lambda_1, \lambda_2),$$

$$\lambda_2 \sim \Gamma(\rho_1, \rho_2).$$

with hyperparameters $\lambda_1$, $\rho_1$ and $\rho_2$ (while this notation is overloaded, we expect it is clear from context which parameter is meant). We model the vector of effect sizes $\boldsymbol{b}_j$ as a multivariate normal, whose mean and covariance is affected by annotation matrices. For gene $j$ we assume annotation matrix $\boldsymbol{F}^j$ and a directed continuous annotation matrix $\boldsymbol{V}^j$, with

dimensions $m_j \times q$ and $m_j \times s$ respectively, with the current implementation of *BAGEA* expecting $\boldsymbol{F^j}$ to be $0 - 1$ coded due to performance reasons. Then the $i$th element of $\boldsymbol{b}_j$ is modeled as

$$b_{ij} \sim N((\boldsymbol{\omega}^T \boldsymbol{v}_i^j)(\boldsymbol{v}^T \boldsymbol{f}_i^j), \alpha_{ji}^{-1}), \tag{7}$$

with $\boldsymbol{v}_i^j$ and $\boldsymbol{f}_i^j$ being the $i$th row of $\boldsymbol{V^j}$ and $\boldsymbol{F^j}$ respectively. $\alpha_{ij}$ being an element of a vector of independently drawn gamma distributed random variables (the independence is conditional on its parental hyperparameters, the modeling of which is described further down). $\boldsymbol{\omega}$ and $\boldsymbol{v}$ are $s$ and $q$ dimensional multivariate normal distributed random variables respectively. $\boldsymbol{\omega}$ denotes the vector of activities of directed annotations, whereas $\boldsymbol{v}$ allows the overall weight that the directed annotations contribute to the effect size vary based on undirected annotations. This allows, for instance, the impact of the directed annotations to vary dependent on the distance to the *TSS*. $\boldsymbol{\omega}$ is modeled in a hierarchical fashion

$$\boldsymbol{\omega} \sim N_s(\boldsymbol{0}, diag(\boldsymbol{\delta}^{-1})),$$

where $\boldsymbol{\delta}$ is again modeled as a random variable. The choice of model for $\boldsymbol{\delta}$ enables the implementation of a grouping structure on the directional annotations (in our application, these groupings are the assay used to derive the annotation and the cell type in which the assay was performed). We allow the model to fit differences in prior variances based on group membership. Thereby, entire groups of directional annotation effects are shrunk to zero (akin to the group lasso [19]). Let $\boldsymbol{d^l}$ be a positive integer vector of length $s$ taking $h_l$ different values, i.e $\boldsymbol{d^l}$ partitions the vector of directed annotations into $h_l$ groups ($l = 1, .., w$ runs over the meta-annotations, e.g. if the modeled meta-annotations are cell type and assay type, $l$ can either take the value one or two). Let $\boldsymbol{v^l}$ be a random vector of length $h_l$ (i.e. these are the group specific weights). Then,

$$\delta_i = \prod_{l=1}^{w} v_{d_i^l}^l,$$

$$v_k^l = \Gamma(\chi_{1l}, \chi_{2l}),$$

with hyperparameter $\chi_{1l}$. $\chi_{2l}$ is modeled as

$$\chi_{2j} \sim \Gamma(\zeta_1, \zeta_2),$$

with hyperparameters $\zeta_1$ and $\zeta_2$.
$\boldsymbol{v}$ is modeled as

$$\boldsymbol{v} \sim N_q(\boldsymbol{c}, diag(\boldsymbol{p})^{-1}),$$

where $\boldsymbol{p}$ and $\boldsymbol{c}$ are hyperparameter vectors of length $q$.
The vector of precisions of the effect size vector $\boldsymbol{\alpha}_j$ is modeled as

$$\alpha_{ij} \sim \Gamma(\gamma_1, \kappa_j \gamma_{ij}),$$

where $\gamma_1$ is a hyperparameter. Note that letting the precision for each SNP vary leads to sparse estimates for $\boldsymbol{b}_j$; this is akin to automatic relevance determination (ARD) regression [18]. $\kappa_j$ is a genewise parameter modeled in a hierarchical fashion

$$\kappa_j \sim \Gamma(\tau_1, \tau_2),$$

$$\tau_2 \sim \Gamma(\xi_1, \xi_2),$$

where $\tau_1$, $\xi_1$ and $\xi_2$ are hyperparameters. To model $\gamma_{ij}$, we again make use of annotation matrices. For gene $j$, assume undirected $0 - 1$ coded annotation matrix $\boldsymbol{C^j}$ of dimension $m \times t$ (BAGEA currently only $0 - 1$ coded matrices for $\boldsymbol{C^j}$ coded due to performance reasons). Then the SNP-wise precision modifier $\gamma_{ij}$ is modeled as

$$\gamma_{ij} = \prod_{k:C^j_{ik}=1} a_k$$

where $\boldsymbol{C^j_{ik}} = 1$ if annotation $k$ is active at index $i$ in gene region $j$. Further,

$$a_k = \Gamma(\phi_1, \phi_2),$$

where $\phi_1$ and $\phi_2$ are hyperparameters.

## Summary statistics adaptation

Instead of using individual level genotype and expression data, we can reformulate the model for the use of summary statistics. Multiplying Eq 6 with $\frac{1}{\sqrt{n}}\boldsymbol{X}^T$ gives

$$\frac{1}{\sqrt{n}}\boldsymbol{X}^T\boldsymbol{y}_j = \frac{1}{\sqrt{n}}\boldsymbol{X}^T\boldsymbol{X}_j\boldsymbol{b}_j + \frac{1}{\sqrt{n}}\boldsymbol{X}^T\boldsymbol{\epsilon}_j.$$

A natural model to use with summary statistics is therefore,

$$\boldsymbol{z}_j = \sqrt{n}\boldsymbol{\Sigma}_j\boldsymbol{b}_j + \boldsymbol{\epsilon}'_j,$$

where $\boldsymbol{z}_j$ is the vector of summary statistics, $\boldsymbol{\Sigma}_j$ is the LD matrix and $\boldsymbol{\epsilon}'_j \sim N_m(\boldsymbol{0}, \lambda_j^{-1}\boldsymbol{\Sigma}_j)$. $\boldsymbol{\Sigma}_j$ can be approximated from external sources such as 1KG [31]. Alternatively, we can use an approximate and regularized version of the empirical LD matrix (see below).

## Model fitting

The model was fit using a variational bayes approach [48]. As the model is in the conjugate exponential family, we can use the variational message passing strategy [49]. For detailed updating steps see S1 Appendix. Naive updates can be prohibitively expensive due to the requirement to invert many large matrices of the form $(c\boldsymbol{X}^T\boldsymbol{X} + \boldsymbol{D_\alpha})$, where $c$ is a constant and $\boldsymbol{D_\alpha}$ is a diagonal matrix. To speed up computation, we can approximate the LD matrix $c\boldsymbol{X}^T\boldsymbol{X}$ with a low rank approximation $\boldsymbol{A}_t^T\boldsymbol{A}_t$, where $\boldsymbol{A_t}$ is a $t \times m$ matrix with $t < m$. This allows us to speed up a time critical matrix inversion step.

$$(c\boldsymbol{X}^T\boldsymbol{X} + \boldsymbol{D_\alpha})^{-1} \approx \boldsymbol{D_\alpha}^{-1} - \boldsymbol{D_\alpha}^{-1}\boldsymbol{A}_t(\boldsymbol{I}_t + \boldsymbol{A}_t^T\boldsymbol{D_\alpha}^{-1}\boldsymbol{A}_t)^{-1}\boldsymbol{A}_t^T\boldsymbol{D_\alpha}^{-1}.$$

If $\boldsymbol{X}$ is already low rank, it is computationally advantageous to use an $\boldsymbol{A_t}$ s.t. $c\boldsymbol{X}^T\boldsymbol{X} = \boldsymbol{A}_t^T\boldsymbol{A}_t$. If $\boldsymbol{A}_t^T\boldsymbol{A}_t$ deviates from $c\boldsymbol{X}^T\boldsymbol{X}$, we need to use the summary statistics formulation to avoid convergence issues. For more detail, see S1 Appendix.

## Deriving annotations

For common SNPs (minor allele frequency (MAF) above 2.5% in the 1000 Genomes European population [31]), we ran the *ExPecto* model to predict the effect of the variant on epigenetic marks [7]. For each SNP we predicted the epigenetic effects within the 200 bp region encompassing it. For most SNPs the effects are very close to zero, allowing us to sparsify the results. Absolute effects smaller than 0.008 were set to zero and all other effects were shrunk towards zero by 0.008 via $x_{new} = x - 0.008 \cdot sgn(x)$. Next, results for both strands were averaged and the

shrinking procedure repeated with a threshold of 0.008. This yielded a matrix with 98.4% of entries zero. The directed annotations were then scaled to have all the same 2-norm. The magnitude of the 2-norm was set to the average of the unscaled 2-norms. These were the directed annotations used in *BAGEA*.

For undirected annotations, we used upstream and downstream distances to the *TSS*. Distance to *TSS* annotations as well as SNP positional annotations were downloaded from the UCSC genome annotation database with SNP and gene annotations taken from the *refGene* and *snp147Common* tables respectively (see link below) [50].

### cis-eQTL datasets

For monocyte eQTL data, we used two preprocessed monocyte datasets with a combined sample size of 1176 (418 from *Fairfax et al.* and 758 from *Rotival et al.* respectively) [20, 21]. Expression matrices were quantile normalized and 10 PEER factors as well as 5 genotype PCs removed [51]. Genotype data was quality control filtered (4% SNP level missingness; 5% individual level missingness; Hardy-Weinberg p-value above $10^{-13}$ relatedness below 0.1875) and imputed using the human genome reference panel [52].

We further downloaded eQTL summary statistics for various tissues produced by the *GTEx* project if the number of samples was above 300 individuals [9]. Additionally, for LCL, we meta-analyzed eQTL summary statistics released for 117 samples by *GTEx* with summary statistics derived from 358 European PEER-controlled samples collected as part of the *GEUVADIS* study [17].

### Running *BAGEA*

For the monocyte eQTL analysis, *BAGEA* was run with default hyperparameter settings (see S1 Appendix). Genotypes within a window of 150KB around a gene's *TSS* were used to construct a genewise LD matrix. Each genewise LD matrix was approximated via singular value decompostion with a low rank symmetric matrix of equal top eigenvalues and eigenvectors, such that the trace of the approximation matrix was at least 99% of the trace of the original LD matrix. Then, a scaled identity matrix was added such that the trace of the resulting matrix was equal to the trace of the original LD matrix. As undirected annotations, distance windows around the TSS (50KB, 20KB, 10KB, 5KB, 2KB, 1KB, 0.5KB, 0.25KB) split into upstream and downstream windows were used. To analyse summary statistics with *BAGEA*, LD was approximated via 1KG European samples. Variants where the reference allele in 1KG did not agree with the reference allele in the UCSC SNP annotation table, were removed. Reference 1KG LD matrices were calculated and replaced with low rank approximations with 95% of the matrix trace kept, anlagously to the above procedure. For all GTEx summary statistics analysis, default hyperparameter settings where used except for **c** which was set to **0.3** instead of **0** to yield consistently positive signs for *v* estimates. *BAGEA* was run for 300 iterations in each analysis.

### URLs

- Code to run *BAGEA* can be found at https://github.com/dlampart/bagea

- Auxiliary preprocessed data automatically installed by *BAGEA* is downloaded from https://s3-us-west-1.amazonaws.com/bagea-data/bagea_data_freeze/

Links to publicly available external data sources are as follows:

- UCSC: http://hgdownload.cse.ucsc.edu/goldenpath/hg19/database/

- ExPecto: https://github.com/FunctionLab/ExPecto/

- 1KG: ftp://ftp-trace.ncbi.nih.gov/1000genomes/ftp/release/20130502/

- GTEX: https://gtexportal.org/home/datasets

- GEUVADIS: ftp://ftp.ebi.ac.uk/pub/databases/microarray/data/experiment/GEUV/
  E-GEUV-3/analysis_results/

- TRRUST-db: https://www.grnpedia.org/trrust/

- Protein atlas: https://www.proteinatlas.org

- First monocyte study: https://www.ebi.ac.uk/ega/studies/EGAS00001000411

- Second monocyte study: https://www.ebi.ac.uk/ega/studies/EGAS00000000109

## Supporting information

**S1 Appendix. Supporting methods.**
(PDF)

**S1 Fig. Removing distance modifier leads to lower predictive power.** We used *BAGEA* to predict gene expression for *CD14* positive monocytes using the *Blood* annotation set, analogously to Fig 2 but removing the distance modifier by constraining all elements of $\hat{\boldsymbol{\nu}}$ except the intercept element to zero (*Without Distance Modifier*). For comparison, we additionally plotted results achieved with the same data and settings except using the default prior for $\boldsymbol{\nu}$ (*With Distance Modifier*). We see substantial decrease in power when the distance dependence of the effect sizes is not modeled.
(TIFF)

**S2 Fig. *BAGEA* SNP effect size estimates predict gene expression out of sample.** SNP effect size estimates $\hat{\boldsymbol{b}}_j$ were derived from a subsample (134 individuals) of one dataset [20]. These estimates were used to predict gene expression in the other available monocyte samples [20] [21]. Fits were performed for genes on chromosomes 1 to 22 that had at least a marginal eQTL p-value of $10^{-10}$ or below in the combined data. Shown is the average estimated expression variance explained in the test data using the *Blood* annotation subset and default distance annotations. *BAGEA* was run either with the default parameter setting (*BAGEA With Annotations*), or with annotation uniformed setting where $\boldsymbol{a}$ was constrained close to $\boldsymbol{1}$ and $\boldsymbol{\omega}$ was constrained close to $\boldsymbol{0}$ (*BAGEA Without Annotations*). Additionally, we compared those estimates to estimates of average genetic variance explained in *cis* as estimated by Haseman-Elston regression on the test data. 95% confidence intervals were derived by bootstrap sampling genes. We see that 60% of estimated genetic variance in *cis* is explained by *BAGEA* out-of-sample estimates of $\hat{\boldsymbol{b}}_j$. Further, running *BAGEA* in annotation uninformed mode dropped this fraction to 0.567%. Overall, we saw that 62.5% of assayed genes had a lower MSE in the annotation informed model than in the annotation uninformed model.
(TIFF)

**S3 Fig. Parameter estimates for the directed annotation TF subset when using *BAGEA* on monocyte eQTL data.** Shown are parameter estimates from fitting monocyte eQTL data using *TF ExPecto* predictions in all cell types. (A) *BAGEA* reveals the experiments underlying the directed annotations that are most predictive of gene expression. **Assay Type x Cell Type**: Each experiment is a particular assay type performed in a particular cell type. **Effect Size** ($\hat{\omega}_i$,

for experiment i): The *BAGEA*-estimated effect on gene expression. Shown here the ten largest directed annotation effect sizes. We see *c-Myc* annotation in *NB4* dominates. (B) Shown is the estimated **distance modifier** of the directed component, $F\hat{v}$. We see a characteristic peak around the *TSS*, implying that the directed annotations are upweighted close to the *TSS*.
(TIFF)

**S4 Fig. Directed annotation effect estimates and modeling error are robust to source of LD information.** (A) Shown is a comparison of estimates of the directed annotation effect vector $\boldsymbol{\omega}$ when using external reference LD information or individual level genotypes. We retrained *BAGEA* with the blood monocyte summary statistics using reference LD matrices from the 1000 Genomes Project (1KG). $\hat{\omega}_i$ (**1KG**): Directed annotation effect, measured as $\boldsymbol{\omega}$ estimates from *BAGEA* using 1KG reference LD information. $\hat{\omega}_i$ (**Monocyte LD**): Directed annotation effect, measured as $\boldsymbol{\omega}$ estimates from *BAGEA* using individual-level genotypes from the monocyte data itself (i.e. using the same genotypes as for the deriving the summary statistics). (B) To investigate the extent to which $MSE_j^{dir}$ can be approximated using summary statistics and reference 1KG LD matrices, we calculated $MSE_j^{dir}$ on chromosomes 16 to 22 from summary statistics of monocyte cis-eQTLs (see formula in main text). We then compared these to the original $MSE_j^{dir}$ values that were computed using genotypes of the monocyte datasets. The same SNPs were used in both calculations. $R^2$: The coefficient of determination, measuring goodness-of-fit, from a linear regression of the data shown.
(TIFF)

**S5 Fig. Simulation results confirm *BAGEA*'s ability to recover relevant annotations.** Shown are precision and recall for various simulation settings (see S1 Appendix) and two parameter settings. For each simulation setting we fitted *BAGEA* either making use of the meta-annotations available for cell type and assay type, (*group-lasso*) or ignoring the meta-annotation information and letting each $\omega_i$ parameter be controlled individual $v_i$ parameter (*lasso*). Upper panels: shown are example results when fitting *BAGEA* either in *group-lasso* setting (A) or *lasso* setting (B), in the *structured* simulation setting with 9 variables (see S1 Appendix for simulation details). True effect sizes for $\boldsymbol{\omega}$ are indicated via red dots. Scaled *BAGEA* estimates of $\boldsymbol{\omega}$ are given as black lines (We scaled $\boldsymbol{\omega}$ to account for differences in estimates of $\hat{v}$ versus $v$. We multiplied each element of $v$ by the coverage of its associated annotations and summed the resulting vector. We treated the estimate $v$ analogously and divided the two to get the scaling factor for $\boldsymbol{\omega}$. These scaling factors where 0.83 and 0.90 for the *group-lasso* (A) and *lasso* (B) setting respectively). When defining all scaled effect size estimates above 0.01 as positives and below 0.01 as negatives, we see that both settings yield a precision of one, whereas *group-lasso* also had a precision of 1.0 and recall of 0.88 and *lasso* had a precision of 0.83 and recall of 0.55 (five out of nine annotations recovered, one false positive). When looking at precision (C) and recall (D) across all simulation settings, we see that precision and recall drop as more variable are added. As expected, in an unstructured simulation setting, it is disadvantageous to enforce a structure on the estimates via the *group-lasso* setting. On the other hand, *group-lasso* maintains good precision and recall in a structured setting with up to 12 variables.
(TIFF)

**S6 Fig. Predictive power of *BAGEA* for various simulation settings show little deviations form predictive power achieved for true parameter settings.** Shown are average $MSE_j^{dir}$ values for all genes in the test set (chromosomes 3). The Upper panel shows average $MSE_j^{dir}$ across all test genes, whereas the lower panel shows average $MSE_j^{dir}$ for genes in the top quartile in terms of $S_j$. We see that the performance is very close to optimal even for settings where

*BAGEA* did not select the correct variables, suggesting that the selected variables were highly correlated to the correct variables.
(TIFF)

**S7 Fig. Comparing *BAGEA* (TF) results with ExPecto result for GTEx data shows favourable performance for genes with large predicted effect sizes.** Shown is the comparison between *BAGEA* and ExPecto for 13 the GTEx experiments w.r.t. agreement between gene expression and the estimated directed predictor. $(S_j > \mathbf{quantile}\ \boldsymbol{x}\ ((S_j > F_{S_j}^{-1}(\boldsymbol{x}))))$: Per GTEx experiments, genes were sorted by the squared magnitude $S_j$ ($S_j$ were computed for both *BAGEA* and ExPecto separately, i.e. for ExPecto $S_j^{ExP}$ was used). For each GTEx experiment, the top $n$-th percent of genes w.r.t $S_j$ were then used to calculate the proportion of genes with positive $\boldsymbol{y}_j^T \hat{\boldsymbol{\mu}}_j$. **(Proportion with $\boldsymbol{y}_j^T \hat{\boldsymbol{\mu}}_j$)**: The proportion of genes for which the scalar product between the gene expression vector $\boldsymbol{y}_j$ and the directed predictor $\hat{\boldsymbol{\mu}}_j$ was larger than 0. We see that for genes with large relative effect sizes, *BAGEA* leads to higher concordance between $\boldsymbol{y}_j$ and $\hat{\boldsymbol{\mu}}_j$. 95% confidence band is derived by bootstrap sampling.
(TIFF)

**S8 Fig. Comparing *BAGEA* (TF) results with Torus result for GTEx data shows comparable performance.** Shown is the comparison between *BAGEA* and *Torus* for 13 the GTEx experiments. To evaluate a method, we determined for each gene in the test set the SNP with the highest prior of being causal. For *Torus*, this amounted to ranking SNPs based on the scalar product between the SNP's annotations and their estimated effect sizes. For *BAGEA*, we ranked SNPs based on $E[b_{ij}^2|\mathbf{G}]$, where $\mathbf{G}$ refers to all global *BAGEA* parameter estimates (see S1 Appendix). **(fitting method)**: The various methods used in the fitting and evaluation. For *Torus* we used various $l_2$ parameter settings as well as an overfitting strategy as upper bound (see S1 Appendix for details) [14, 15]. **(Proportion with $|z_{top}| - |z_{annot}| < 0.2$)**: To evaluate a given method, we picked the SNP for each gene in the test set for which the method predicted the largest absolute effect sizes based on the annotations alone and recorded its z-score (denoted $|z_{annot}|$). We then compared this value to the overall largest absolute z-score for this gene (denoted $|z_{top}|$). We evaluated the power by the proportion of genes for which $|z_{top}| - |z_{annot}|$ was below 0.2. 95% confidence interval is derived by bootstrap sampling.
(TIFF)

**S9 Fig. Comparison of effect size estimates between *BAGEA* and Torus.** Shown are the distribution of effect size estimates of *Torus* when fitted on 13 GTEx datasets for the *TF* annotation subset ($l_2 = 100$). As *Torus* was run in ridge mode, few effects were very close to zero. *BAGEA* in its default parameter resembles lasso, in that it only selects a limited number of effect sizes substantially different from zero. When fitting *BAGEA* using the same datasets in *lasso* mode, we saw 257 annotations overall larger than 0.001 (of which 192 where also larger than 0.01). When color-coding those 257 effect sizes based on direction and comparing them against the rest, we saw that the *Torus* effect sizes were markedly shifted away from zero for both the positive and the negative effect size *BAGEA* group.
(TIFF)

**S10 Fig. Histogram of directed effect sizes $\hat{\boldsymbol{\omega}}$ across all 14 GTEx datasets.** Displayed are estimated directed annotation effect sizes $\hat{\omega}$ for all GTEx (and GEAUVADIS) datasets, with values with absolute value below $10^{-3}$ removed. Shown are results when fitting on data from all autosomes.
(TIFF)

**S11 Fig. Largest negative directed annotation effect sizes for GTEx summary statistics repeat some of the same tissue pairings as large positive effect sizes.** Shown are the largest negative directed annotation effect from fitting 14 different GTEx (and GEAUVADIS) eQTL summary statistics datasets using Histone and DHS ExPecto predictions derived from Roadmap.
(TIFF)

**S12 Fig. *BAGEA* fits for GTEx summary statistics with non-histone ChIP-seq annotations show largest effect sizes for Pol2 assays.** Shown is the largest directed annotation effect for each of the fitted 14 different GTEx (and GEAUVADIS) eQTL summary statistics datasets using ExPecto predictions derived from ENCODE non-histone ChIP-seq experiments [33].
(TIFF)

**S13 Fig. *BAGEA* fits with TF annotations predict TF activators and repressors.** After removal of *Pol2* from the *TF* annotations subset, we fit *BAGEA* to the GTEx summary statistics using *lasso* mode. (A) Shown are the number of GTEx experiments for which a given TF-ChIP-Seq assay shows a postively or negatively signed effect with absolute value above 0.01 (If multiple annotations mapped to the same TF we summed the effects, this step only affected a few TFs because of the regularization strategy employed). (B) Shown is the comparison between *BAGEA*'s prediction of repressor/activator activity of a TF's with predictions derived from the trrust-db v2 [46]. **(sign bias of effect size direction (#))**: For each TF we take the difference between the number of postive effect directions (blue bar in panel (A)) and the number of negative effect directions (red bar in panel (A)) to get a prediction of whether a TF acts as activator ($>0$) or repressor ($<0$). **(Fraction of Repressor Annotation (TRRUST-db))**: The fraction of annotations in the human TTRUST db (human) for a given TF which claimed repressor activity among all annotations with a clear assigned direction (i.e. after removal of all annotations with unknown direction from TRRUST-db). We see a clear dependence between results from TRRUST (unidirectional p-value lower than 0.0001, $R^2 = 0.43$), suggesting that results from *BAGEA* are in broad agreement with the literature in terms of determining activator and repressor TFs.
(TIFF)

**S14 Fig. Comparison of distance modifier estimates for *BAGEA* fits on GTEx data.** Shown is the estimated **distance modifier** of the directed component, $F\hat{v}$ for all GTEx experiments when fit with the *TF* annotation subset in *lasso* mode. Individual results are plotted in grey and averages are plotted in black. While there is some fluctuation for individual results around the mean, the general peak shape is respected in all cases.
(TIFF)

**S15 Fig. Speed of variable update varies across annotation subset.** Shown is the speed with wich each updating iteration of the variational algorithm takes for the main analyses performed. For *Blood* we used the setting of the monocyte analysis (see for instance Fig 2). For *TF* and *Histone/DHS* we used the settings used in the respective GTEx analyses (see for instance Fig 4). All analyses were performed on an *AWS r*4 × 4 instance using 15 cores. As we ran the algorithm for 300 iterations, we see that in this setting, the algorithm took between 50 minutes and 5 and a half hours to complete.
(TIFF)

**S1 Table. BAGEA effect size estimates for GTEx experiments.** Given are the $\omega$ effect size estimates for various *BAGEA* fits to GTEx data. Only effect sizes with absolute value above

0.001 are included.
(TXT)

**S2 Table. BAGEA effect size estimates for monocyte experiment.** Given are the $\omega$ effect size estimates for various *BAGEA* fits to the monocyte data. Only effect sizes with absolute value above 0.001 are included.
(TXT)

**S3 Table. Directed Annotations to tissue/cell type mapping.** Given are the mappings between the ExPecto annotations and the Roadmap tissues, as well as ENCODE cell lines.
(TXT)

**S4 Table. $MSE^{dir}$ estimates for GTEx experiments.** Given are the $MSE^{dir}$ on the test set for various *BAGEA* fits to the GTEx data.
(TXT)

**S5 Table. $MSE^{dir}$ estimates for monocyte experiments.** Given are the $MSE^{dir}$ on the test set for various *BAGEA* fits to the monocyte data.
(TXT)

## Acknowledgments

Special thanks to Prof. Zoltan Kutalik for helpful discussions.

## Author Contributions

**Conceptualization:** David Lamparter, Rajat Bhatnagar, Katja Hebestreit, T. Grant Belgard, Victor Hanson-Smith.

**Formal analysis:** David Lamparter.

**Funding acquisition:** Alice Zhang.

**Methodology:** David Lamparter.

**Project administration:** Victor Hanson-Smith.

**Software:** David Lamparter.

**Writing – original draft:** David Lamparter, Rajat Bhatnagar, Katja Hebestreit, T. Grant Belgard, Alice Zhang, Victor Hanson-Smith.

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
