## [Decision Letter · Decision Letter 0]

11 Sep 2019

Dear Dr Hanson-Smith,

Thank you very much for submitting your manuscript 'A framework for integrating directed and undirected annotations to build explanatory models of cis-eQTL data' for review by PLOS Computational Biology. Your manuscript has been fully evaluated by the PLOS Computational Biology editorial team and in this case also by independent peer reviewers. The reviewers appreciated the attention to an important problem, but raised some substantial concerns about the manuscript as it currently stands. While your manuscript cannot be accepted in its present form, we are willing to consider a revised version in which the issues raised by the reviewers have been adequately addressed. We cannot, of course, promise publication at that time.

Sincerely,

Roger Pique-Regi, Ph.D.

Guest Editor

PLOS Computational Biology

Weixiong Zhang

Deputy Editor

PLOS Computational Biology

[LINK]

This is an important and interesting problem, however all three reviewers agree that the authors should explain better what gain in accuracy and in biological understanding do the directed annotations provides. To this end, the authors should address this by using a more clear mathematical notation and provide a more accurate explanation and intuition for their modeling choices. Additional literature covering this problem should also be covered, and additional methods should also be used for comparison. These comparisons using simulated and/or real datasets suggested by the reviewers should also further highlight any advantage, if any, the directed annotations can provide in terms of accuracy or gained biological insight. The reviewers also note that the code needs to be improved and all the data be made available.

Reviewer's Responses to Questions

**Comments to the Authors:**

Reviewer #1: Summary

- This paper describes a framework to understand gene expression in different genomic datasets (e.g. tissue type). This framework can use sequence-specific directed and undirected annotations. It can predict what types of genomic annotations drive gene expression in different conditions.

- The method is tested using multiple eQTL datasets, Expecto mutation predictions as directed annotations, and distance from TSS as undirected annotations. It is tested with a full gene expression and genotype dataset as well as with summary statistics and estimated linkage matrices. The annotations that it prioritizes appear to be biologically relevant.

- The novelty of the method is that it is the first to use directed annotations to determine the types of annotations that are important in different genomic datasets, and that it can weigh the contribution of these directed annotations based on distance to TSS. Some previous methods can predict directed effects of mutations, and others can determine what types of undirected annotations drive gene expression in a dataset, including distance to TSS. But none does both.

- The framework is interesting and perhaps important for the future of the field, but this paper does not offer any comparison to existing undirected annotation tools. The lack of comparison makes it difficult to determine the utility of using directed annotations, especially at the higher computational burden. It also makes it difficult to determine how much biological insight the framework can add to the field.

- The current framework may offer significant methodological insight and enable future researchers to build on its use of directed annotations. However, the lack of comparison with existing methods presents a serious problem, and the biological findings reported do not represent any significant biological insight.

Major issues

1. I can imagine the utility of using directed annotations to interpret gene expression, but I don’t think that utility has been shown. The method prioritizes certain genomic annotations that make biological sense. But there’s no comparison to the annotations that are prioritized by existing tools that use undirected annotations only. There’s no comparison to a simple enrichment test of genomic annotations of eQTL variants, or a more sophisticated enrichment test using e.g. DAP/torus. The introduction also does not include a discussion of current approaches that use undirected annotations.

2. This paper appears to rely heavily on the Expecto algorithm (ref 7). Expecto is used to generate the directed annotations by calculating the difference in predicted epigenetic signals [e.g. DNase, histone marks] between the reference and alternative alleles of a SNP. These values are then entered into the BAGEA directed annotation matrix. Expecto can already use those values to predict the effect of a given mutation on gene expression (although I don’t think it takes TSS distance into account). The current method’s utility is that using the Expecto annotations, it can learn which of the annotations are relevant for genetic effects in a given dataset on a genome-wide basis. While this seems like it could be useful and interesting, it’s difficult to judge without a metric for comparison (major issue #1).

3. The prioritized annotations per dataset do not appear to offer major biological insight. The findings provided are a good “sanity check” that the expected annotations are prioritized, and there are a couple minor biological insights (e.g. bone-marrow-derived mesenchymal stem cells vs. fibroblasts, DNase1 peaks vs. hotspots). It would add to the paper if the methodology could be used to create new hypotheses about a new or unknown dataset or biological question.

Minor issues

1. The authors should provide more context for Sj/MSEdir and highlight the expected values possible based on the cis genetic varition. What is the maximum expression variance that their method could have been expected to explain? The finding that “6.6% of the total genetic variance in cis was explained by the externally-fitted directed component mu_j for genes in the top quartile” helped me understand the usefulness and biological relevance of the method. I’m also curious how much genetic variance was explained by only the variants that end up being used for prediction, as 98.4% of the entries in the directed matrix were set to zero.

2. The method introduces an intuitive way to learn the pattern of the importance of distance to TSS. However, I wonder how much this pattern varies between genes and between datasets, and I don’t see any data on this. If the pattern doesn’t vary much, as I imagine it might not, why not just keep Fv fixed instead of learned?

3. It would be interesting to know if there are differences between genes with low MSE and high MSE. Is there anything about the high MSE genes that makes it easier to predict their gene expression using this model?

4. There are no comments about the finding that some traditionally activating annotations (e.g. Fig 3C CD3 DNase peaks) have negative effect sizes for certain datasets. I can speculate myself, but I think it would be worth discussing.

5. The introduction begins by explaining the importance of understanding the effects of rare variants, but the method as implemented only uses common variants (MAF > 2.5% in EUR).

6. It would be useful to provide a list of all the Roadmap annotation used (or at least the cell types), so that readers know which annotations were not prioritized. Perhaps a full list of Figure 5A could be included in the supplement.

7. Figure 1B seems more complicated than necessary. Maybe a grid would be a better tool for representation.

8. It is difficult to follow the trend lines and see the different layers in Figures 2A and 4B, especially in Figure 4B.

9. Including the tissue sample size or number of eQTLs per tissue might add more meaning to Figure 5A.

10. Since it looks like there are a couple outliers, it might be more informative to sort the x-axis of Figure 5B by median instead of by maximum. And it would be interesting to know which tissues those outliers came from.

11. It appears the GTEx data were analyzed using a EUR-only LD matrix, when the sample is ~85% EUR.

12. The method does not use INDEL information (I believe).

13. I tried to download and test the software myself, but the install script would not work; it failed on “devtools::use_rcpp()” with the message “Error: 'use_rcpp' is not an exported object from 'namespace:devtools'”. I did not investigate further.

Reviewer #2: The authors present a model connecting genomic annotations, predicted effects of variants on those annotations, and eQTLs. The key innovation is the the model can consider both directed and undirected annotations: in particular variant effect predictions (e.g. from deep learning models) and distance from TSS respectively. The model is constructed and initially tested for the scenario where genotypes and gene expression are available, but an extension to using summary statistics and reference panel LD is presented that seems to recapiluate the full data results well. The underlying model is a "soft sparse" Bayesian regression with ARD prior on coefficients (this gives a student-t distribution on each coefficient, the authors should check that and mention it i so). The annotations are incorporated into the mean of the prior. The vector of directed annotation a SNP, v, is dot producted with a learnt coefficient vector w. Similarly, the undirected annotations f are dot producted with nu. These two scalars are multiplied together to give the prior mean for the SNP effect on expression (I think it might be helpful to give this explanation in the text since the matrix version is a bit more obtuse. The prior variance can optionally be stochastically dependent on a third binary annotation matrix C. Priors (and hyperpriors) are setup on all variables and VBEM is used for learning. The an affine transform of the "LD matrix" X'X in principle needs inverting each iteration, which is costly. They pre-compute a low rank approximation (explained e.g. 99% of variance) and then use the matrix inversion lemma subsequently.

The paper is very clearly presented. There are small number of typos I've annotated on the attached manuscript. I wish PLOS wouldn't tell authors to upload low quality bitmaps and put the figures at the end, but that's a review of the journal not the paper (for future reference they will let you submit a pdf with inline figs at first submission).

The model itself is very reasonable. The main aspect one could argue about is whether this approach of "soft sparsity" is competitive with spike-and-slab priors as used in Bayesian fine-mapping approaches such as enloc or Matt Stephen's recent SuSiE method.

The results themselves are of course not overwhelming, with only a small fraction of genetic variance (let alone total variance) explained by the model, even for the most well predicted genes. I would be curious to know what are the reason(s) driving this: variant effector predictors aren't good enough? insufficient sample size for training BAGEA (seems unlikely)? Chromatin state is in only proxy cell types?

While the small pve is what it is, I would appreciate some attempt to benchmark vs existing eQTL/annotation approaches like eQTNMiner, enloc or stratified LD score regression. I realize this is non-trivial since existing approaches can't predict gene expression, and there is no ground truth for what annotations should be important for eQTLs in a given cell type. That leaves simulations, which are likely to bias towards which model is most similar to the simulation, and fine mapping. Fine mapping might be a reasonable way to compare (although I realize the method is not specifically tailored to this). You could think of downsampling individuals and see how well you recover the credible set of variants from the full data.

One thing I would emphasize is that this model could be applied to very rare, even de novo, variants. This distinguishes the approach from both existing eQTL and eQTL+annotation methods.

Maybe I missed it but do you have a run time analysis? This is relevant for practitioners.

Is there a good reason the undirected annotations are binary? The math looks like it would all go through just fine with non-negative annotations.

Reviewer #3: Lamparter et al. propose a new computational method, BAGEA, to model cis-eQTL data using genomic annotations. They argue that the new computational model can identify epigenetic marks relevant to expression biology. For the most part, the paper is clearly written. However, I have some concerns about some critical model assumptions, evaluations of the proposed model, and the analysis of the real eQTL data.

Major comments:

1. Model assumptions. There is generally a lack of discussions on the intuition and motivation of modeling decisions. First, a unique perspective of the proposed method is its ability to model *directed genomic annotations*. However, it is unclear how important are the *directions* of the annotations enforced in the model. In particular, are the omega parameters constrained (e.g., in the motivating example of binding affinity?). If yes, how? If not, I fail to understand how the concept of directed genomic annotations differs from a quantitative annotation. Relating to the same point, should the variance term alpha in Eqn (2) depend on the mean term too? Because a potentially large independent variance term can make the directional effect meaningless. Second, the joint modeling of directed and undirected annotations in Eqn (2) is not intuitive. It is unclear to me why it is a reasonable strategy that the model *only* consider the interactions between the two distinct categories. The particular model formulation also causes confusions: consider a single directed annotation and a single undirected annotation, would the absence of the undirected annotation (i.e., 0) completely remove the effect of the quantitative information from the directed annotation?

2. Evaluation of the model. The authors choose to evaluate the proposed model based on its predictive performance. However, their results show extremely inaccurate predictions at the absolute scales (1.5% variance explained). It is well known that gene expressions are generally difficult to predict, but other available genotype predictive approaches, e.g., the simple elastic net algorithm implemented in PrediXcan method, show much higher heritability based on similar datasets (See https://www.ncbi.nlm.nih.gov/pubmed/27835642). Can the authors explain the discrepancy? On the surface, this seems to undermine the main point of the paper: by introducing an annotation-informed prior, I'd expect at least similar, if not much better, predictive performance (an annotation-uninformed model should be included as a special case of the proposed model by setting appropriate nu, omega and alpha parameters).

3.The relevance of genomic annotations. More details should be provided on this topic. Although some computational difficulty is acknowledged in the discussion, the adopted training and testing protocol should be evaluated in a simulation setting. It is also vague from the paper if an individual annotation is assessed marginally or under the control of other competing annotations.

Minor comments:

1. undirected annotations are not formally defined as directed annotations.

2. The GitHub page for the software needs to be improved. It does not have proper documentation or example datasets. I strongly encourage the authors to provide the datasets (i.e., the summary statistics from eQTL data and genomic annotations) analyzed in this paper online for the reproducibility purpose.

**Have all data underlying the figures and results presented in the manuscript been provided?**

Reviewer #1: No: I did not receive a spreadsheet form of the numerical data underlying graphs/summary statistics.

Reviewer #2: Yes

Reviewer #3: None

PLOS authors have the option to publish the peer review history of their article (what does this mean?). If published, this will include your full peer review and any attached files.

Reviewer #1: No

Reviewer #2: Yes: David A Knowles

Reviewer #3: No

---

## [Decision Letter · Decision Letter 1]

30 Jan 2020

Dear Dr. Hanson-Smith,

Thank you very much for submitting your manuscript "A framework for integrating directed and undirected annotations to build explanatory models of cis-eQTL data" for consideration at PLOS Computational Biology. As with all papers reviewed by the journal, your manuscript was reviewed by members of the editorial board and by several independent reviewers. The reviewers appreciated the attention to an important topic. Based on the reviews, we are likely to accept this manuscript for publication, providing that you modify the manuscript according to the review recommendations.

Please make sure to address the remaining minor comments from Reviewer 1. As well as asking for clarification on some further issues, they have highlighted a few spelling mistakes/typos - please also take this opportunity to read through carefully for any others. 

Sincerely,

Roger Pique-Regi, Ph.D.

Guest Editor

PLOS Computational Biology

Weixiong Zhang

Deputy Editor

PLOS Computational Biology

[LINK]

Reviewer's Responses to Questions

**Comments to the Authors:**

Reviewer #1: Thank you to the authors for thoroughly addressing the previous comments and adding the requested analyses.

A few new, minor comments:

1. In "Joint modeling of cis-eQTLs and Directed Annotations..." you mentioned a cell line rationale for the prioritized TF annotations. Were the expected TFs also prioritized, or did the effects appear to be mostly driven by cell lines?

2. Line 352, "annotation" sp.

3. Was torus run using all eQTLs (all p-values)? Or were the same variants used for both methods (p<10e-7)?

4. Could comment #3 possibly explain why BAGEA showed no effect for many non-zero torus effects (Fig S9)? It's also possible that I'm misunderstanding the text -- I'm not entirely sure that I understand the last sentence of the paragraph (lines 360/361).

5. Fig 4. caption partially sp.

6. Fig 4A: What's going on with Adipose Omentum -- why is the RM MSE at 1? Apologies if I missed this discussion in the text.

7. When discussing DNase1 data processing, it might be helpful to mention that DNase1 hotspots are usually wider than DNase1 peaks calls (I believe).

Reviewer #2: I've read the other reviews, author feedback and changes. I was already quite positive about the paper but I think with the additions now made it should certainly be accepted.

**Have all data underlying the figures and results presented in the manuscript been provided?**

Reviewer #1: Yes

Reviewer #2: None

PLOS authors have the option to publish the peer review history of their article (what does this mean?). If published, this will include your full peer review and any attached files.

Reviewer #1: No

Reviewer #2: Yes: David A Knowles
---

## [Editor Report · Decision Letter 2]

3 Mar 2020

Dear Dr. Hanson-Smith,

We are pleased to inform you that your manuscript 'A framework for integrating directed and undirected annotations to build explanatory models of cis-eQTL data' has been provisionally accepted for publication in PLOS Computational Biology.

Best regards,

Roger Pique-Regi, Ph.D.

Guest Editor

PLOS Computational Biology

Weixiong Zhang

Deputy Editor

PLOS Computational Biology

---

## [Editor Report · Acceptance letter]

29 Apr 2020

PCOMPBIOL-D-19-01078R2 

A framework for integrating directed and undirected annotations to build explanatory models of cis-eQTL data

Dear Dr Hanson-Smith,

I am pleased to inform you that your manuscript has been formally accepted for publication in PLOS Computational Biology. Your manuscript is now with our production department and you will be notified of the publication date in due course.

With kind regards,

Bailey Hanna
